# Kinetics of blood cell differentiation during hematopoiesis revealed by quantitative long-term live imaging

Kevin Yueh Lin Ho[1†], Rosalyn Leigh Carr[1,2,3†], Alexandra Dmitria Dvoskin[1], Guy Tanentzapf[1]*

[1]Department of Cellular and Physiological Sciences, University of British Columbia, Vancouver, Canada; [2]School of Biomedical Engineering, University of British Columbia, Vancouver, Canada; [3]British Columbia Children's Hospital, Vancouver, Canada

**Abstract** Stem cells typically reside in a specialized physical and biochemical environment that facilitates regulation of their behavior. For this reason, stem cells are ideally studied in contexts that maintain this precisely constructed microenvironment while still allowing for live imaging. Here, we describe a long-term organ culture and imaging strategy for hematopoiesis in flies that takes advantage of powerful genetic and transgenic tools available in this system. We find that fly blood progenitors undergo symmetric cell divisions and that their division is both linked to cell size and is spatially oriented. Using quantitative imaging to simultaneously track markers for stemness and differentiation in progenitors, we identify two types of differentiation that exhibit distinct kinetics. Moreover, we find that infection-induced activation of hematopoiesis occurs through modulation of the kinetics of cell differentiation. Overall, our results show that even subtle shifts in proliferation and differentiation kinetics can have large and aggregate effects to transform blood progenitors from a quiescent to an activated state.

**\*For correspondence:**
tanentz@mail.ubc.ca

[†]These authors contributed equally to this work

**Competing interest:** The authors declare that no competing interests exist.

## Editor's evaluation

This study represents an important technical advancement in the live-imaging and analytical approaches to the *Drosophila* larval hematopoietic organ called the lymph gland. The new method allows tracking the proliferation and differentiation of progenitor cells ex vivo and provides insights into the modes of differentiation during development and infection. The evidence supporting this is convincing but would be further strengthened if the authors explain the disparity between the speed and proportion of two modes of differentiation as the reviewers suggested.

## Introduction

Because of their key role in tissue maintenance and the inherent risk associated with their unchecked proliferative capacity, stem cell behavior is tightly regulated (*Klein and Simons, 2011*; *He et al., 2009*). This regulation is often mediated by controlling the biochemical composition of the microenvironment surrounding the stem cells, including the presence of signaling molecules and metabolic cues (*Jones and Wagers, 2008*; *Morrison and Spradling, 2008*). Additionally, physical inputs, such as mechanical cues and the architecture and topography of the extracellular matrix, also play an important role in controlling stem cell behaviour (*Chacón-Martínez et al., 2018*; *Ahmed and Ffrench-Constant, 2016*; *Díaz-Torres et al., 2021*). The complex and precisely constructed in vivo microenvironment of stem cells can be very challenging to mimic in the laboratory and, when possible, it is best

to study stem cells in their endogenous environment. There has been significant emphasis over the last decade on the development and optimization of imaging tools and methodologies to allow live imaging of stem cells in their in vivo environment (*Park et al., 2016*; *Rompolas et al., 2012*; *Martin et al., 2018*; *Sheng and Matunis, 2011*).

Hematopoiesis, the production of the cellular components of blood, is a well-known paradigm for stem cell regulation through a niche (*Martinez-Agosto et al., 2007*; *Huang et al., 2007*; *Mandal et al., 2007*; *Tokusumi et al., 2010*). *Drosophila* provides a powerful and genetically tractable model to study hematopoiesis (*Banerjee et al., 2019*). During fly hematopoiesis, a specialized population of blood progenitors gives rise to blood cells. *Drosophila* blood progenitors exhibit many stem-cell-like properties, such as being controlled by a specialized population of cells that act as a hematopoietic niche, the Posterior Signaling Centre (PSC)(*Krzemień et al., 2007*, *Mandal et al., 2007*; *Tokusumi et al., 2010*; *Cho et al., 2020*). They can give rise to three highly differentiated blood cell types: plasmatocytes, lamellocytes, and crystal cells (*Jung et al., 2005*). However, whether they are true stem cells has not been conclusively resolved (*Banerjee et al., 2019*). Vertebrate hematopoietic stem cells (HSCs) have been associated with several well-defined attributes: self-renewal, the ability to differentiate into all blood lineages, and their dependence on a niche (*Huang et al., 2007*). *Drosophila* blood progenitors have been shown to exhibit most of these criteria but so far, there has been no clear evidence of either self-renewal or asymmetric cell division in the progenitors (*Banerjee et al., 2019*). Nevertheless, previous studies provided evidence for the existence of HSCs in the lymph gland (LG) (*Cho et al., 2020*; *Dey et al., 2016*; *Minakhina and Steward, 2010*). For example, a population of cells in first instar larvae were identified, which gave rise to progenitors in later larval stages and behaved in ways consistent with the idea that they were equivalent to vertebrate HSCs (*Dey et al., 2016*). Another study employed MARCM-based lineage tracing technique to identify distinct subpopulations of progenitors within third instar larvae, one of which exhibited characteristics such as 'persistence' that were consistent with the HSC fate (*Minakhina and Steward, 2010*).

The main site of hematopoiesis in *Drosophila* larvae is the primary lobe of a specialized organ known as the LG (*Banerjee et al., 2019*). The primary lobe contains three distinct zones: the PSC niche, the medullary zone (MZ) which houses the blood progenitors, and the cortical zone (CZ) that contains differentiated blood cells (*Banerjee et al., 2019*; *Mandal et al., 2007*). The progenitors in the MZ express markers such as the JAK-STAT receptor Domeless (Dome), while differentiated blood cells express unique markers depending on their terminal mature blood cell fate. For example, plasmatocytes express the marker P1, while crystal cells express the marker Lozenge (Lz) (*Banerjee et al., 2019*; *Jung et al., 2005*). In addition, a small population of cells has been described that lack the expression of terminal differentiation markers such as P1 but express both the progenitor marker Domeless and early differentiating blood cell markers such as Hemolectin (Hml) and Peroxidasin (Pxn) (*Blanco-Obregon et al., 2020*). These P1⁻, Dome⁺, and Pxn⁺/Hml⁺ cells were typically found in the boundary between the MZ and CZ and were proposed to represent a separate population of cells, commonly referred to as intermediate progenitors (*Blanco-Obregon et al., 2020*; *Sinenko et al., 2009*; *Spratford et al., 2021*; *Girard et al., 2021*). Although this population is currently not well characterised, it is thought to contain cells in a transitional state as they go from a relatively quiescent multipotent state to a terminally differentiated state (*Krzemien et al., 2010*; *Banerjee et al., 2019*).

More recent data has supported the view that rather than being a homogenous population, progenitors in the MZ are heterogeneous (*Blanco-Obregon et al., 2020*; *Cho et al., 2020*; *Baldeosingh et al., 2018*; *Girard et al., 2021*). Single-cell transcriptomic analysis of the LG revealed a surprising level of heterogeneity of the developing blood cells and uncovered novel blood cell types including adipohemocytes, stem-cell like blood progenitors, and intermediate progenitors (*Cho et al., 2020*). Moreover, a distinct subpopulation of progenitor cells in the LG has been recently identified and termed 'distal progenitors' (*Blanco-Obregon et al., 2020*). These cells, named after their location at the distal part of the MZ near the boundary with the CZ, express some progenitor markers (Dome) but not others (Tep4) (*Blanco-Obregon et al., 2020*). A further subpopulation of distal progenitors, known as 'committed progenitors' is distinguished by its expression of the plasmatocyte marker gene *eater* but not the mature blood cell marker Hml (*Blanco-Obregon et al., 2020*). The population of progenitors that are close to the heart tube additionally exhibit distinct features in terms of regulation by Hh signalling, a key regulator of blood cell differentiation in the LG (*Baldeosingh et al., 2018*). These data suggest that rather than being composed of a simple and clearly defined population of

progenitors, the LG contains multiple subpopulations of progenitors at various stages and states of differentiation. These observations show the limitations of using fixed tissue approaches to study blood progenitor fate which is inherently a dynamic and evolving cell state.

The main function of mature blood cells in *Drosophila* is to fight infection and assist in wound healing (*Evans et al., 2003*; *Khadilkar et al., 2017*; *Vlisidou and Wood, 2015*). It is therefore not surprising that in healthy intact flies, few mature blood cells are made in late larval stages (*Banerjee et al., 2019*). During early larval stages, blood progenitors undergo expansion and are typically found to be in S phase of the cell cycle (*Dey et al., 2016*). However, once progenitor expansion is completed in the late larval stages, they for the most part stay in the G2 phase through the action of dopamine (*Sharma et al., 2019*; *Kapoor et al., 2022*), although some intermediate progenitors remain in S phase (*Sharma et al., 2019*). Upon infection, there is a strong and rapid induction of mature blood cell production, which depends on the type of immune challenge and involves large scale differentiation of lamellocytes, crystal cells, or plasmatocytes (*Banerjee et al., 2019*; *Khadilkar et al., 2017*; *Letourneau et al., 2016*). How this induction is mediated, for example, whether it is primarily driven by changes in the cell cycle in the progenitors, predominantly by altered dynamics or patterns of differentiation, or by both factors in equal measure, is currently unclear. The key to answering these important remaining questions is to develop the ability to visualize and track fly hematopoiesis for extended periods in real time.

Here, we describe analysis of proliferation and differentiation patterns observed during long-term live imaging of intact LGs in healthy and infected larvae. This analysis utilises whole organ culture methodology and quantitative imaging tools that we developed, optimised, verified, and applied. By tracking markers for cell proliferation and division in real time as well as cell fate and differentiation, we are able to confirm that blood progenitors undergo symmetric cell divisions. Using quantitative automated image analysis of progenitors in healthy and infected flies, we elucidate the dynamics and spatiotemporal patterns of blood cell differentiation and proliferation. We describe how the modulation of key differentiation and proliferation behaviours underlies the activation of mature blood cell production following infection. These results provide a novel system-level framework for understanding how *Drosophila* hematopoiesis is regulated in the context of the intact whole organ in real time.

## Results

### Development and optimization of a long-term ex vivo whole organ LG culture and imaging technique

In order to image fly hematopoiesis in real time, we developed a whole organ culture system for the LG. A large number of protocols for culturing various organs were explored and, through trial and error, we found that optimal results were obtained with a modified version of protocols used for imaging whole testes, CNS, and wing imaginal discs (*Fairchild et al., 2015*; *Zartman et al., 2013*; *Reilein et al., 2018*; *Sheng and Matunis, 2011*; *Anllo et al., 2019*; *Tsao et al., 2016*; *Martin et al., 2018*; *Kakanj et al., 2020*; *Morris and Spradling, 2011*; *Kiepas et al., 2020*; *Icha et al., 2017*; *Greenspan and Matunis, 2017*; *Zhang et al., 2018*; *Dai et al., 2020*; *Koyama et al., 2020*). This method used Schneider's cell culture medium and relied upon three key features that we found to dramatically improve outcomes: (1) dissection methodology, wherein the LG was removed while maintaining its association with the CNS, ring gland and the heart tube with which it was then co-cultured, (2) the addition of intact larval fat bodies to the culture, and (3) the use of spacers to prevent mechanical force from being applied to the tissue by the presence of the cover slip and agar pad (see Materials and methods). With this technique, LG ultrastructure and proliferative capacity were maintained, and it was found we could successfully culture LGs overnight (*Figure 1A–D*, *Figure 1—figure supplement 1*, *Video 1*). We used a genetically encoded marker for oxidative stress, gstD-GFP (*Sykiotis and Bohmann, 2008*), to show that oxidative stress in the LG does not increase substantially over the course of 13 hrs of ex vivo culture and imaging (*Figure 1E*). As a control and to illustrate the ability of gstD-GFP to detect oxidative damage, we omitted the fat bodies from the culture medium and observed a large increase in oxidative stress in the LGs over time (*Figure 1E*; *Figure 1—figure supplement 2A–B*). Furthermore, direct comparison of LGs that were kept in ex vivo culture conditions and physiological in vivo conditions over the course of 12 hours showed comparable levels of

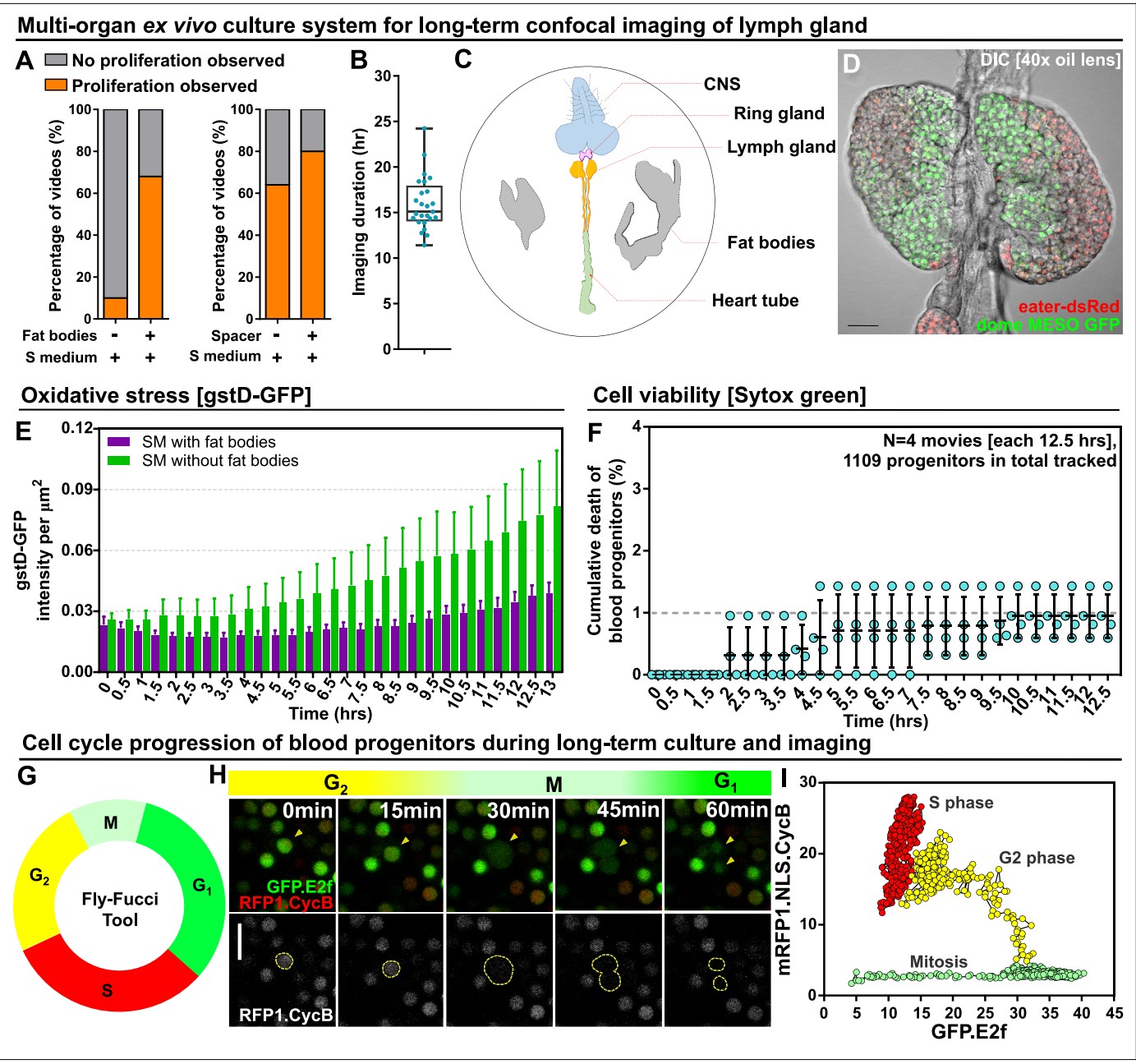

**Figure 1.** Long-term ex vivo culture system for extended imaging of developing Lgs. (**A**) Quantification of the percentage of videos where proliferation was observed versus videos where no proliferation was observed under four different conditions: co-culture with or without the presence of fat bodies (percentage calculated from n=29 videos) and with or without placing a spacer (percentage calculated from n=19 videos). (**B**) Quantification of the duration of imaging (in hrs) for individual LG videos (n=25 videos, on average 15.96 hr per video; see Materials and methods). (**C**) Schematic showing the experimental setup for the multi-organ co-culture system in a glass bottom dish. Organs in the culture include the central nervous system (CNS), ring gland, LG, heart tube (or dorsal vessel), and fat bodies. (**D**) Representative DIC image of an ex vivo cultured LG (blood progenitors labelled with dome-MESO-GFP in green, mature hemocytes labelled with eater-dsRed in red). Genotype of the LG was *dome-MESO-GFP; eater-dsRed*. (**E**) Quantification of oxidative stress levels in whole LGs cultured overnight under two conditions: in Schneider's medium (SM) with fat bodies (n=13 primary lobes tracked from 8 videos [each 13 hr]) and in SM without fat bodies (n=7 primary lobes tracked from 4 videos [each 13 hr]). Genotype of the LG was *gstD-GFP*. (**F**) Quantification of blood progenitor viability during long-term live imaging. In total n=1109 progenitors (marked by Tep4-Gal4 driven dsRed) were tracked from 4 videos (each 12.5 hr). Genotype of the LG was *Tep4-Gal4>UAS-dsRed*. (**G**) Schematic of the Fly-FUCCI system used to track the cell cycle progression using distinct fluorescent markers in combinations (see Materials and methods). (**H**) An example showing G2 to M to G1 transition of a blood progenitor over the course of approximately 60 min. (**I**) Quantification from an example, using Tep4-Gal4 driven FUCCI, to visualize an S to G2 to M progression of a blood progenitor. Each dot represents a time point; decrease in the intensity during mitosis was caused by nucleus breakdown. Genotype of the LG was *Tep4-Gal4>UAS-FUCCI*. Scale bars in (**D**) and (**H**) represent 50 and 10 μm, respectively. Error bars indicate S.D from the

*Figure 1 continued on next page*

*Figure 1 continued*

mean. S medium in (**A**) and SM in (**E**) denote Schneider's medium supplied with 15% FBS and 0.2 mg/mL insulin (see Materials and methods). See also *Videos 1–3*.

The online version of this article includes the following source data and figure supplement(s) for figure 1:

**Source data 1.** Raw data of *Figure 1A, B, E, F, I*.

**Figure supplement 1.** A LG stays integrated during long-term culture and imaging.

**Figure supplement 2.** Long-term ex vivo cultured LGs exhibit low oxidative stress and high cell viability.

**Figure supplement 2—source data 1.** Raw data of *Figure 1—figure supplement 2A, B, C, E*.

**Figure supplement 3.** Ex vivo cultured LGs demonstrate comparable cell cycle, proliferation, and differentiation profiles to in vivo LGs.

**Figure supplement 3—source data 1.** Raw data of *Figure 1—figure supplement 3A, B, C, D, E, F, G*.

oxidative stress (*Figure 1—figure supplement 2C*). Similarly, the cell death stain Sytox green was used to monitor cell viability (*Martin et al., 2018*) and showed that, while there is a very small baseline level of cell death in the progenitors and in the whole LG, this baseline did not increase in a substantial way over the course of LG culture (*Figure 1F*; *Figure 1—figure supplement 2D–E*, *Video 2*). As a control and to illustrate the ability of Sytox green to monitor cell viability, we cultured LGs in PBS instead of Schneider's medium in the presence of fat bodies, which led to a marked increase in cell death over time (*Figure 1—figure supplement 2E* right panel). Moreover, the baseline level of cumulative cell death that we observed under ex vivo culture conditions during 12.5 hr of culture was 2–12 cells per LG (*Figure 1—figure supplement 2E* left panel) which was in line with what was observed in previous in vivo studies (0–10 cells)(*Khadilkar et al., 2020*; *Araki et al., 2019*; *Yu et al., 2021*; *Chiu and Govind, 2002*; *Mondal et al., 2011*). Overall, we did not detect any harm or damage to the LG caused by the ex vivo culture technique.

To test and demonstrate the ability of our culture system to allow tracking of cell behaviour in the LGs over extended periods of time, we utilised the Fly-FUCCI system (*Zielke et al., 2014*) to monitor cell cycle progression in the progenitors. Fly-FUCCI is based on the expression of fluorescent protein-tagged degrons from the Cyclin B and E2F1 proteins, which are degraded during

**Video 1.** Long-term imaging of an ex vivo wild-type LG at single cell resolution. Representative long-term live imaging video of a primary lobe from an ex vivo LG showing blood progenitor divisions (highlighted by yellow ROIs in the video) over a cultured period of 13 hr. Blood progenitors were marked by dome-Gal4-driven membranous GFP (green). Mature hemocytes were marked by eater-dsRed (red). Part of the ring gland (labelled as RG in the video) was also captured. The LG was obtained from an early 3rd instar larva (of genotype *dome-Gal4 >UAS-mCD8-GFP, eater-dsRed*) raised at 25 °C, dissected, immediately mounted and imaged. Scale bar: 10 µm.

https://elifesciences.org/articles/84085/figures#video1

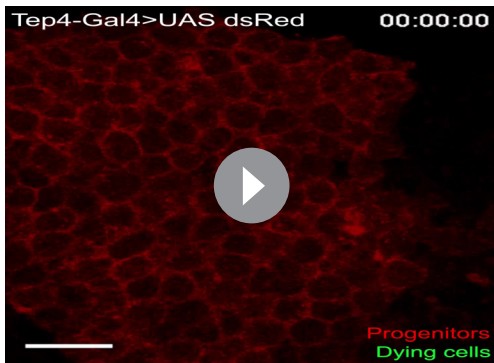

**Video 2.** Long-term monitoring of blood progenitor viability during overnight ex vivo culture. Representative video showing only three blood progenitors undergoing cell death in a live LG cultured ex vivo over a period of 12 hr. The blood progenitors were marked by Tep4-Gal4-driven dsRed (red). Dying cells were marked by Sytox green dye (green). The LG was obtained from an early 3rd instar larva (of genotype *Tep4-Gal4>UAS-dsRed*) raised at 25 °C, dissected, immediately mounted and imaged. Scale Bar: 15 µm.

https://elifesciences.org/articles/84085/figures#video2

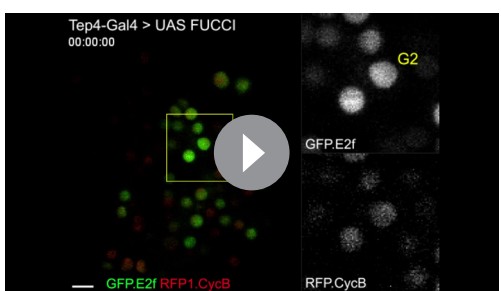

**Video 3.** Long-term tracking of cell cycle progression of blood progenitors in a wild-type LG. Representative video showing cell cycle progression of a blood progenitor (highlighted in a yellow ROI) in wild-type LG over a period of 1 hr. The cell cycle indicator FUCCI construct was expressed in blood progenitors using Tep4-Gal4. The video tracked a G2-M transition (G2 phase: have both GFP.E2f and RFP.CycB expressed) of a blood progenitor and the subsequent G1 progenies (G1: only have GFP.E2f expressed). The green and red channels were separately presented in the right side of the video to visualize GFP.E2f and RFP.CycB levels in individual blood progenitors over time. The LG was obtained from an early 3rd instar larva (of genotype *Tep4-Gal4>UAS-FUCCI*) raised at 25 °C, dissected, immediately mounted and imaged. Scale Bar: 10 μm.
https://elifesciences.org/articles/84085/figures#video3

mitosis or the onset of S phase, to distinguish the G1, S, and G2 phases of interphase (*Zielke et al., 2014*). Expressing Fly-FUCCI using a progenitor-specific driver (Tep4-Gal4) allowed us to track in real-time the cell cycle phase of individual progenitors using colour as an indicator: green during G1, red during S phase, and yellow during G2 (*Figure 1G–H*; *Video 3*). We applied auto-mated quantitative imaging tools (see Materials and methods) to track the trajectory of a single progenitor through the cell cycle in order to analyse the dynamics of the process (*Figure 1I*). The Fly-FUCCI system allowed us to examine if our ex vivo culture methodology recapitulated the in vivo behaviour of LGs. Specifically, we compared the cell cycle profile of LGs in vivo (in intact stage-matched larvae) and in ex vivo culture. We observed little change in the proportion of cells in the LGs at each stage of the cell cycle over time in ex vivo culture (*Figure 1—figure supplement 3A*), while, if we deliberately stressed LGs by leaving out the spacers and allowing them to be compressed, we saw cell cycle arrest (*Figure 1—figure supplement 3B*). Moreover, tracking the proportion of cells in the LGs at each stage of the cell cycle over time either in ex vivo culture or in vivo showed little difference between the two conditions (*Figure 1—figure supplement 3C*).

Notably, the cell cycle data we obtained using ex vivo cultured LGs showed ~20% of cells were in G2,~40% of cells were in S, and ~40% of cells were in G1, numbers that were similar to previous in vivo observations from mid-third instar larvae (*Rodrigues et al., 2021*). Finally, comparing proliferation in in vivo or ex vivo conditions by EdU labelling showed similar proliferative capacity in both conditions (*Figure 1—figure supplement 3D–E*). Importantly, there was no reduction in proliferation over long-term ex vivo culture and our observation of ~100 EdU[+] cells per primary lobe was in line with previous data collected in fixed LGs (*Milton et al., 2014*). Further evidence for the health of the ex vivo cultured LGs over the course of long-term imaging was provided by noting that the duration of the mitosis remains largely consistent over time (*Figure 1—figure supplement 3F*). Taken together, the data illustrate that we can track the cell cycle and proliferation in the LG using our culture technique and validate our approach as we did not detect any obvious cell cycle defects or variance from published data collected using fixed in vivo conditions (*Rodrigues et al., 2021*; *Milton et al., 2014*).

## Blood progenitors in the LG undergo symmetric cell divisions

A key unresolved question about the fly blood progenitors is whether they undergo self-renewal or symmetric cell divisions (*Banerjee et al., 2019*). We used long-term LG imaging to address this question directly and found multiple lines of evidence that suggest that blood progenitors undergo symmetric cell divisions. Cultured LGs expressing both the JAK-STAT reporter and progenitor marker dome-MESO-GFP (*Oyallon et al., 2016*) as well as the early differentiation marker eater-dsRed (*Kroeger et al., 2012*) were imaged. We observed many examples in multiple videos from different LGs of symmetric cell divisions in progenitors (34 dividing progenitors from 7 videos). Since these progenitors are identified as dome-MESO-GFP expressing cells that do not express eater-dsRed (*Figure 2A*; *Video 4*) they are either core progenitors or distal progenitors (*Figure 2—figure supplement 1A*). These dividing core or distal progenitors maintained their dome-MESO[+] eater-dsRed[-] fate in both daughter cells after cell division. To confirm this observation, we employed another way to label progenitors by using the dome-Gal4 driver line (*Jung et al., 2005*) to drive the expression of membranous GFP (*Figure 2B*; *Video 1*). We found that progenitor divisions are symmetrical with

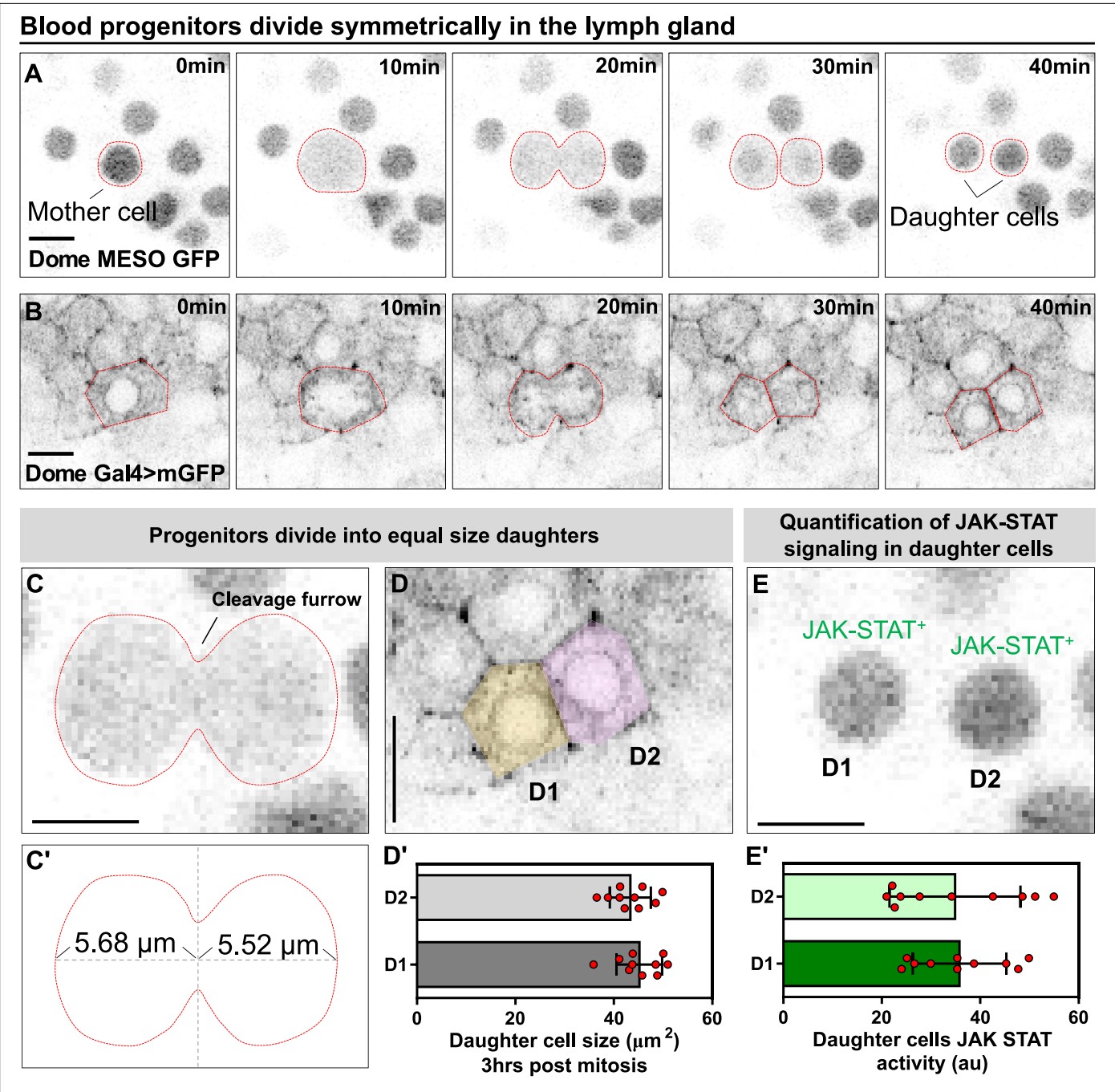

**Figure 2.** Blood progenitors undergo symmetric divisions in the LG. (**A, B**) Time-lapse images from representative videos showing a blood progenitor (labelled as "mother cell") undergoing symmetric division over the course of 40 min (progenies labelled as 'daughter cells'). Blood progenitors were labelled with dome-MESO-GFP (grey) and highlighted with red outline (**A**) or labelled with dome-Gal4 driven membranous GFP (grey) and highlighted with red outline (**B**). (**C-C'**) Representative image of the contractile ring positioned at the cleavage furrow of a dividing blood progenitor (**C**) and a schematic illustrating the distance between the contractile ring to the two poles of the cell (**C'**, μm). (**D-D'**) Representative image (**D**) and quantification (**D'**) of the cell area (μm²) of the two daughter cells 3 hrs post mitosis (n=10 daughter cell pairs randomly pooled from 3 videos). (**E-E'**) A representative image (**E**) and quantification (**E'**) of the JAK-STAT signaling activity (a.u) of the daughter cells (n=10 daughter cell pairs randomly pooled from 7 videos). All scale bars represent 5 μm. The genotype of the LGs shown in (**A**), (**C**), (**E**) was *dome-MESO-GFP; eater-dsRed* and in (**B**), (**D**) was *dome-Gal4 >UAS mGFP; eater-dsRed*. Error bars indicate S.D from the mean. See also ***Video 4***.

The online version of this article includes the following source data and figure supplement(s) for figure 2:

**Source data 1.** Raw data of ***Figure 2D and E***.

**Figure supplement 1.** Genetic tools and markers to study blood progenitor cell fate transition during hematopoiesis.

*Figure 2 continued on next page*

*Figure 2 continued*

**Figure supplement 2.** Long-term tracking of dome-MESO-GFP intensities in daughter progenitor cells.

**Figure supplement 2—source data 1.** Raw data of *Figure 2—figure supplement 2*.

respect to size, with a cleavage plane positioned in such that the cell division yields two daughter cells of similar size (*Figure 2C–C'*; *Figure 2D–D'*). Another hallmark of progenitor fate is a high level of JAK-STAT signalling activity (*Oyallon et al., 2016*). Quantifying the intensity of dome-MESO-GFP in daughter cells as a readout for activity of the JAK-STAT pathway shows that following progenitor division the daughter cells exhibit similar levels of the signalling (*Figure 2E–E'*; see Methods). To account for the possibility that it is due to equal inheritance of the protein from the mother, not an equivalent maintenance of a progenitor fate, these experiments were done in the presence of eater-dsRed to confirm neither of the daughter cells differentiated. Also supporting the equivalent maintenance model, we found that tracking dome-MESO-GFP levels in daughter cells over extended periods of time showed that the marker levels remained stable and did not diverge in both cells (*Figure 2—figure supplement 2*). Taken together, the data provide compelling evidence that blood progenitors in *Drosophila* undergo symmetric division that produces two identical progenitor cells that are conserved in both cell size and JAK-STAT signaling pathway.

## Blood progenitor division is linked to cell size and is spatially oriented

Since we were able to track a large number of dividing progenitors in real time, identified as dome-MESO[+] eater-dsRed[-] cells, in the LG (*Figure 3A–B*), we were able to quantitatively analyse the kinetics of cell growth and division. We found that most dividing progenitors complete cell division in 40–70 min (*Figure 3B*; 57.74±27.58 minutes, n=63 progenitors). It has been shown that the cell division is often coordinated with cell size and can be initiated by cells reaching a so-called 'critical cell size' (*Lengefeld et al., 2021*; *Ferrezuelo et al., 2012*). For dome-positive progenitors, division occurred once cells reached an average size of 72 µm² (*Figure 3C* left panel, 71.96±10.00 µm², n=13 progenitors). Upon cell division, two similar sized progenitors are produced and undergo rapid growth such that their combined size exceeds the size of the original mother cell 3 hr after division, making cell division a potential driver for LG growth (*Figure 3C* right panel). Analysis of the growth kinetics of the two daughter cells over time showed that the two daughter cells can grow 20–30% in the first 4 hr after division (*Figure 3D*). As these experiments focused on the entire progenitor population, we sought to gain more detailed insight by labelling specific sub-populations of progenitors. First, we asked which progenitor sub-populations in the LG were mitotically active by constructing a fly line that carried the following markers: Tep4-QF>QUAS-mCherry, dome-MESO-GFP, and HmlΔ-dsRed. This allowed us to mark core progenitors (Tep4-mcherry[+] dome-MESO-GFP[+]), distal progenitors (only dome-MESO-GFP[+]) and intermediate progenitors (dome-MESO-GFP[+] HmlΔ-dsRed[+]; *Figure 2—figure supplement 1A*; *Figure 3—figure supplement 1A–A'''*). We then used a pH3 staining to determine which of the progenitor populations were mitotically active. We found that core and distal progenitors were mitotically active while intermediate progenitors were not (*Figure 3—figure supplement 1A–A'''*). This finding is consistent with published results showing intermediate progenitors are not mitotically active (*Spratford et al., 2021*). Next, we

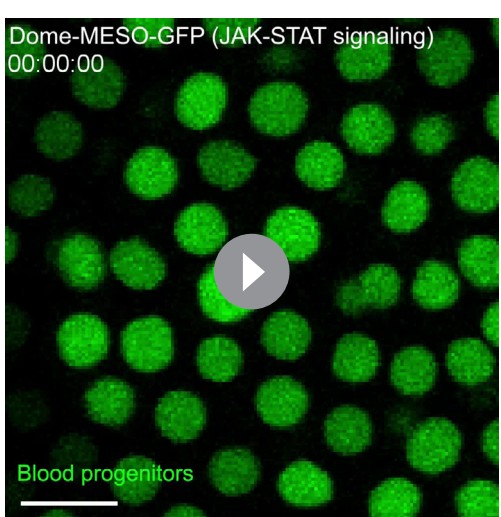

**Video 4.** Blood progenitors divide symmetrically in a wild-type LG. Representative video showing an example of a blood progenitor undergoing symmetric cell division over a period of 50 min (see also *Video 1*). The blood progenitors were marked by JAK-STAT signaling activity reporter dome-MESO-GFP (green). The daughter cells were marked by yellow ROIs. The LG was obtained from an early 3rd instar larva (of genotype *dome-MESO-GFP*) raised at 25 °C, dissected, immediately mounted and imaged. Scale Bar: 10 µm. https://elifesciences.org/articles/84085/figures#video4

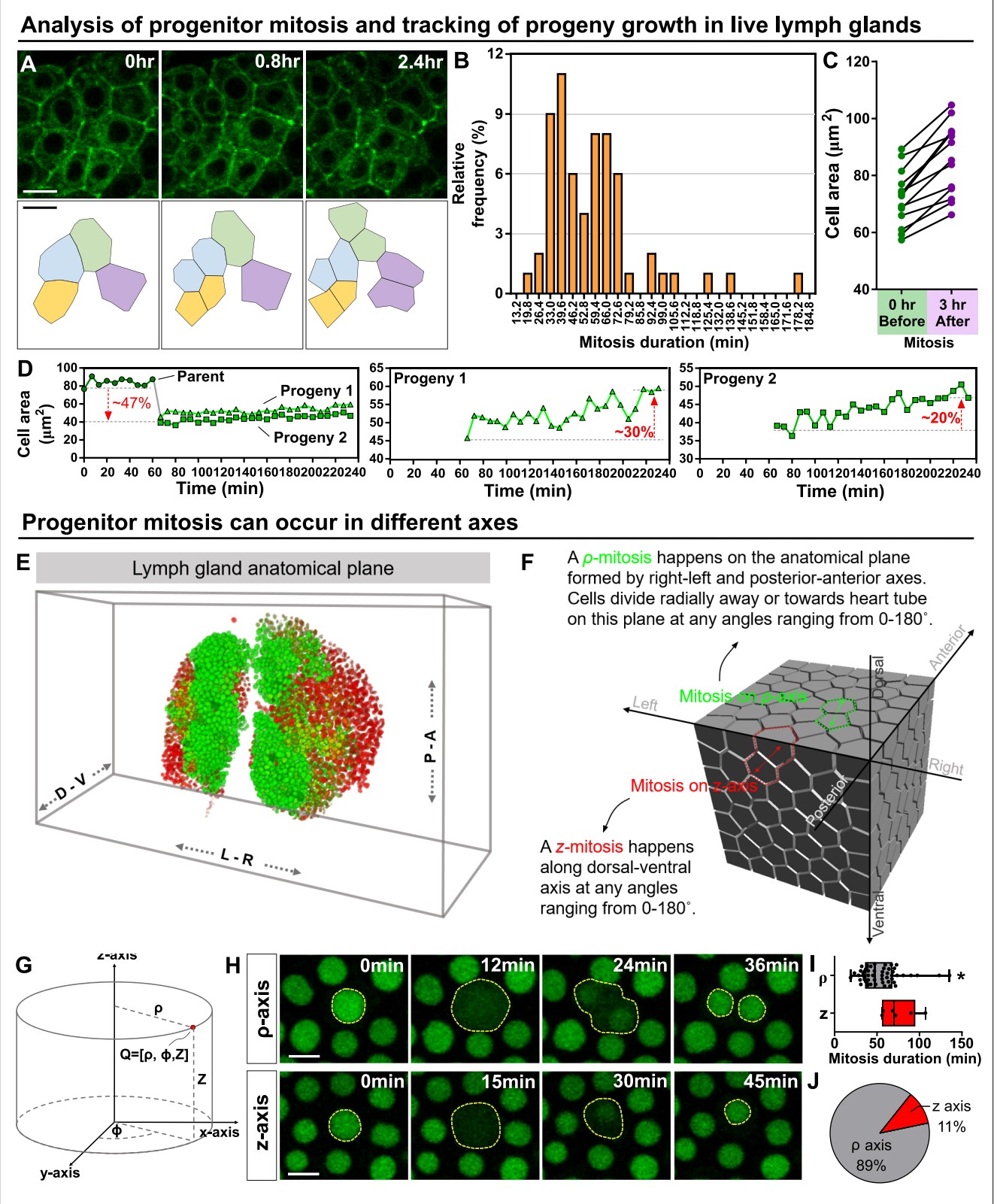

**Figure 3.** Kinetics of blood progenitor mitosis, progeny growth, and mitotic axes in the LG. (**A**) Top panel: time-lapse images from a representative video showing multiple dividing blood progenitors over 2.4 hr. Blood progenitors labelled with dome-Gal4 driven membranous GFP (of genotype *dome-Gal4 >mGFP; eater-dsRed*). Bottom panel: schematic of four dividing blood progenitors and their progenies outlined based on the fluorescent images from the top panel. (**B**) Histogram showing the distribution of the duration of mitosis (in minutes) of blood progenitors (n=63 dividing blood

*Figure 3 continued*

progenitors collected from 10 videos). The majority of blood progenitors spent an average of 33–73 min in mitosis. (**C**) Quantification of the cell area (µm²) of mother blood progenitors 10 min before mitosis (data highlighted in the left panel, each data point represents a mother cell) and cell area summed up from two generated progenies 3 hr after mitosis (data highlighted in the right panel, each data point represents cell area summed up from two generated progenies; n=10 progenitors of genotype *dome-Gal4 >UAS mGFP; eater-dsRed* undergoing mitosis tracked over 3 hr). (**D**) Representative example selected from (**C**) showing the quantification of progeny growth (reflected by their cell area, µm²). Left panel: real-time measurement of the cell area of the parent (or mother) cell, progeny 1, and progeny 2. Middle and right panels: changes in the ell area of progeny 1 and 2 over time, respectively. (**E**) Schematic showing anatomical axes of a 3D LG (A: anterior, P: posterior, L: left, R: right, D: dorsal, V: ventral; blood progenitors marked by dome-MESO-GFP in green, mature hemocytes marked by eater-dsRed in red). The LG is shown following a convention established previously for a 3D representation of the fly CNS (*Zheng et al., 2018*). (**F**) Detailed schematic showing mitotic events happening on the $\rho$ and z axes with respect to the anatomical axes. Concept of $\rho$ and z axes is derived from the cylindrical coordinate system (as shown in G; see Methods). The 3D cell matrix was built using codes from Geogram Delaunay3D. (**G**) Diagram showing the cylindrical coordinate system ($\rho$-, $\phi$-, z-axes) compared to a Cartesian coordinate systems (x-, y-, z-axes). (**H**) Time-lapse images from representative videos of progenitor mitotic events occurring along the $\rho$-axis over 36 min (top panel) or along the z-axis over the course of 45 min (bottom panel). Blood progenitors labelled with dome-MESO-GFP (green, LG genotype: *dome-MESO-GFP; eater-dsRed*). (**I**) Quantification of the durations (in minutes) of blood progenitor mitotic events occurring along the $\rho$-axis (n=54 progenitors) and z-axis (n=6 progenitors). p-value = 0.022 was determined using Mann-Whitney-Wilcoxon test. * indicates p<0.05. (**J**) Pie graph showing the percentage of recorded blood progenitor mitotic events occurring along the $\rho$-axis and z-axis. The data in *Figure 2A, C, E* and (**F–H**) came from the same live imaging experiments but different cells were analysed and presented. The data in *Figure 2B, D* and (**A–D**) came from the same live imaging experiments but different cells were analysed and presented. All scale bars represent 5 µm. Error bars indicate S.D from the mean. See also *Video 4*.

The online version of this article includes the following source data and figure supplement(s) for figure 3:

**Source data 1.** Raw data of *Figure 3B, C, D, I, J*.

**Figure supplement 1.** Critical cell size and spatial distribution of blood progenitor divisions.

**Figure supplement 1—source data 1.** Raw data of *Figure 3—figure supplement 1B, C, D, E, F, G*.

---

found that the critical size for distal progenitors (dome⁺ Tep4⁻) was on average 76.08 µm², while for core progenitors (dome⁺ Tep4⁺) it was on average 63.81 µm² (*Figure 3—figure supplement 1B*). Our analysis shows that distal progenitors have to reach a larger critical cell size than core progenitors before they can initiate mitosis (*Figure 3—figure supplement 1B*). As an internal control for changes in cell size, we selected a neighbouring cell that did not undergo mitosis as a comparison to the mitotic cell analysed (*Figure 3—figure supplement 1B*).

In many niche-stem cell systems, the stem cells exhibit spatial polarisation in regards to the orientation of the dividing stem cells (*Martin et al., 2018*). We considered whether blood progenitor divisions were spatially polarised. In general, we described the anatomical planes of a LG and the polarity of cell division using anatomical axes of the LG coupled in relation to a cylindrical coordinate system (*Figure 3E–G*), which describes organ anatomical structure in greater mathematical simplicity than other coordinate systems by allowing the radial direction from the dorsal-ventral axis to be defined explicitly with the use of trigonometry (*Rood et al., 2019*). The $\rho$-axis of cylindrical coordinate system corresponds to the radial axis, which is parallel to the plane formed by anterior-posterior and right-left axes of a larva and defines how far from the origin a given point lies, the θ-axis defines the absolute angle of the given point from the origin, while the z-axis corresponds to dorsal-ventral axis (*Figure 3E–G*; see Methods) and defines the 'height' of the point on the now defined $\rho$-θ (radial length-angle) plane. This coordinate system was chosen as divisions were found to either always radiate out from the dorsal-ventral axis ($\rho$-mitosis) or run along the dorsal-ventral axis (z-mitosis; *Figure 3F*). Using the cylindrical coordinate system, all $\rho$-mitosis can be compared directly as the angle of the radial axis is a separate coordinate (compared to usual Cartesian coordinates, where radial measurements rely on both the x and y-axes). We found that approximately 90% of the divisions occurred along the $\rho$-axis, and that these divisions took a shorter time to complete (on average 53.90 min for $\rho$-axis divisions versus on average 75.04 min for z-axis divisions; *Figure 3H–J*). Notably, the overall shape of the LG which is 'longer (roughly 300 µm) and wider (roughly 150 µm)' than it is 'thick (roughly 40–60 µm)' is consistent with such a polarised orientation of division. Divisions along the $\rho$-axis tended to occur in dividing progenitors located further away from the heart tube or PSC when compared to divisions along the z-axis (*Figure 3—figure supplement 1C–E*). To statistically confirm that $\rho$-axis divisions are less likely to occur parallel to the heart tube (see wild-type control in *Figure 4L*), we used a quantile-quantile (Q-Q) plot to compare the distribution of division orientation we observed to a randomised normal distribution (*Figure 3—figure supplement 1F*; see Materials

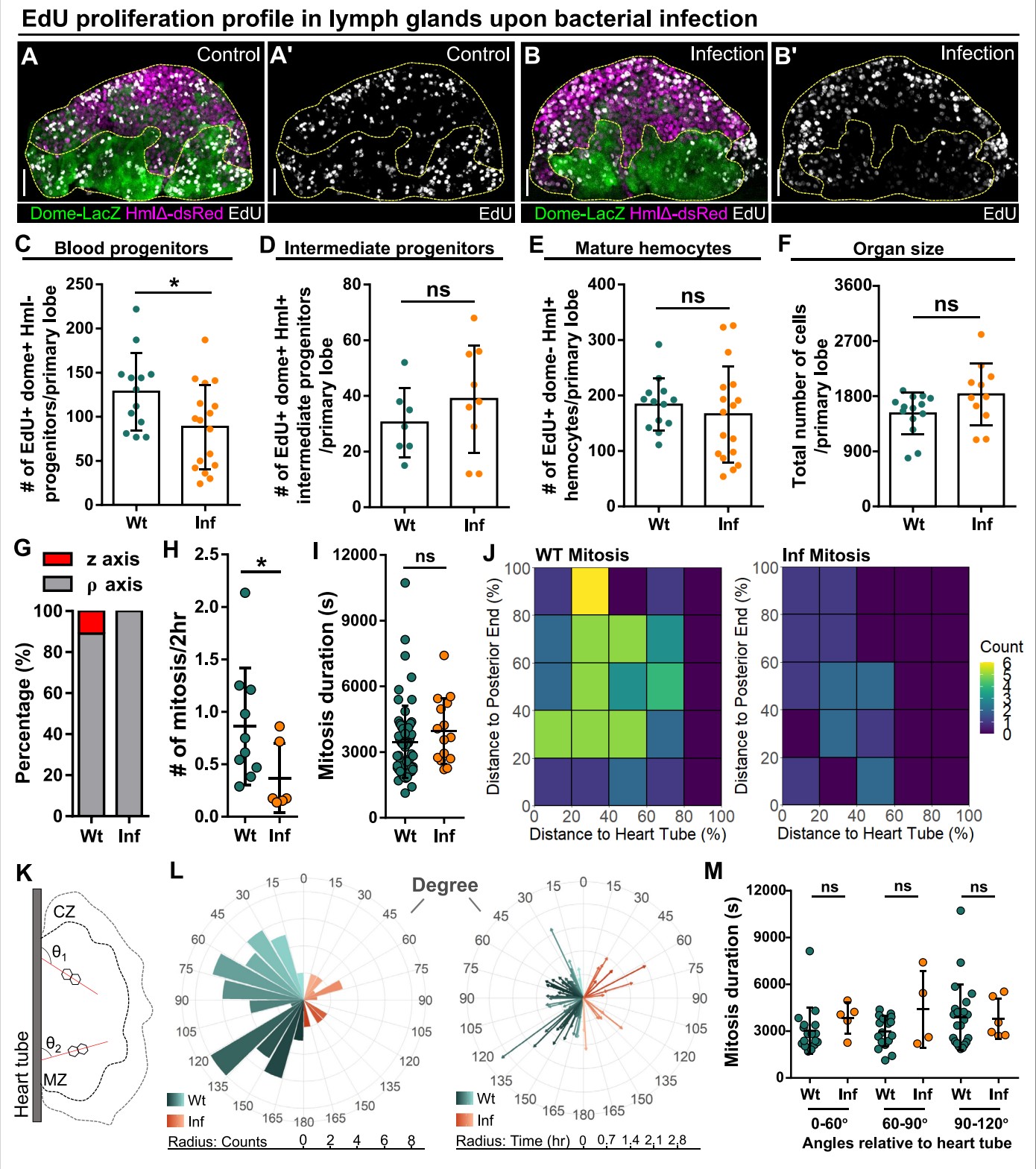

**Figure 4.** Reduction of blood progenitor proliferation upon infection. (**A–E**) Representative images (**A–B**) and quantification of EdU incorporation in blood progenitors (C, dome[+] Hml[-], n=13 lobes in wild-type control, n=17 lobes in infection group, *P*-value = 0.032), intermediate progenitors (D, dome[+] Hml[+], n=13 lobes in wild-type control, n=17 lobes in infection group, p-value = 0.336), and mature hemocytes (E, dome[-] Hml[+], n=7 lobes in wild-type control, n=9 lobes in infection group, p-value = 0.3465) in LGs from wild-type control and *E. coli* infected larvae. Proliferating cells (or cells actively synthesizing DNA) labelled with EdU (white). Progenitors labelled with dome-MESO-LacZ (green). Mature hemocytes labelled with HmlΔ-dsRed

*Figure 4 continued on next page*

*Figure 4 continued*

(magenta). Genotype of the LG was *dome-MESO-LacZ; HmlΔ-dsRed*. (**F**) Quantification of the total number of cells per primary lobe from wild-type control (n=14 lobes) and *E. coli* infected larvae (n=11 lobes). p-value = 0.0872. (**G**) Quantification of the percentage of mitotic events occurring in blood progenitors along either the z-axis or the $\rho$ -axis in LGs from wild-type control and *E. coli* infected larvae. (**H**) Quantification of the number of mitotic events in blood progenitors in LGs from wild-type control (n=10 videos) and *E. coli* infected larvae (n=6 videos). p-value = 0.0397. (**I**) Quantification of the duration of mitotic events in LGs from wild-type control (n=63 dividing progenitors) and *E. coli* infected larvae (n=15 dividing progenitors). p-value = 0.1365. (**J**) Heat maps summarising data from long-term imaging experiments showing the number of mitotic events recorded in distinct regions of LGs from wild-type control and *E. coli* infected larvae (see Materials and methods). (**K**) Schematic illustrating the orientation ( $\theta_1$ and $\theta_2$) of the $\rho$ -mitosis with respect to the heart tube. CZ: cortical zone, MZ: medullary zone. (**L**) Left panel: rose diagram showing the distribution of the orientation of mitotic events (in degree) in LGs from wild-type control and *E. coli* infected larvae. Radius corresponding to the number of mitotic events recorded. Right panel: rose diagram showing the duration (in hrs) of each mitotic event occurring at different orientations from wild-type control and *E. coli* infected larvae. Radius corresponding to the duration of mitotic events. (**M**) Quantification of the duration of progenitor mitotic events occurring at different angles relative to the heart tube (0°–60°, 60°–90°, and 90°–120°) in LGs from wild-type control and *E. coli* infected larvae (p-value = 0.0824, 0.4342, and 0.6331 in 0°–60°, 60°–90°, and 90°–120° groups, respectively). Mitotic progenitors analysed in (**G–M**) were all dome⁺ progenitors from LGs having the following genotype: *dome-MESO-GFP; eater-dsRed*. Scale bars in (**A-A' and B-B'**) represent 40 μm. Error bars indicate S.D from the mean. p Values were determined using Mann-Whitney-Wilcoxon test. ns indicates non-significant, p>0.05. * indicates p<0.05.

The online version of this article includes the following source data and figure supplement(s) for figure 4:

**Source data 1.** Raw data of *Figure 4C, D, E, F, G, H, I, L, M*.

**Figure supplement 1.** Reduction of blood progenitor divisions upon infection.

**Figure supplement 1—source data 1.** Raw data of *Figure 4—figure supplement 1C, D*.

**Figure supplement 2.** Workflow of spatial analysis on cell divisions and differentiations.

and methods). This analysis showed that the orientation of $\rho$ -axis divisions relative to the heart tube is biased or polarised (*Figure 3—figure supplement 1F*). Overall, these observations uncover a previously unknown polarization in progenitor divisions in the LG as these divisions tend to occur more frequently along the plane formed by the anterior-posterior and right-left axes.

## Infection results in reduced cell proliferation in the LG

Following infection, the LG undergoes a dramatic change as the cellular immune response is activated and there is a large-scale induction of differentiation of mature blood cells (*Khadilkar et al., 2017*). Increased production of mature blood cells can be achieved by a number of possible scenarios including: (1) increased proportion of progenitor cells undergoing cell division, (2) no change in the proportion of dividing progenitors but a faster cell cycle, (3) increased proportion of progenitors undergoing differentiation, (4) no change in the proportion of progenitors undergoing differentiation, but faster differentiation, (5) any combination of these options. We analyzed LGs from larvae infected with *E. coli* bacteria using a previously developed infection protocol (see Materials and methods; *Khadilkar et al., 2017*; *Siva-Jothy et al., 2018*). We labelled cell proliferation using EdU (a marker for proliferation; *Figure 4A–E*) or pH3 (a marker for mitosis; *Figure 4—figure supplement 1A–D*) and quantified proliferations and cell divisions in LGs from wild-type control and infected larvae. We simultaneously labelled different cell populations in the LG using dome-MESO-LacZ and HmlΔ-dsRed to identify blood progenitors (dome⁺ Hml⁻), intermediate progenitors (dome⁺ Hml⁺), and mature blood cells (dome⁻ Hml⁺). We observed a reduction in the number of dividing progenitor cells following infection (*Figure 4A–A', B–B' , and C*; *Figure 4—figure supplement 1A–A', 1B-B', and 1C*) and there was no change in either the proliferation of intermediate progenitors (*Figure 4D*) or mature blood cells (*Figure 4E*; *Figure 4—figure supplement 1D*) or in LG size (*Figure 4F*). This data from in vivo fixed tissues was confirmed by quantifying cell divisions in long-term live imaging experiments of dome⁺ progenitors in the LGs from wild-type control and infected larvae which showed a reduction in the number of cell division events (*Figure 4H*). In terms of the duration of mitotic events, there was no significant difference between progenitors in the LGs from wild-type control and infected larvae (*Figure 4I*). There was, however, a slight change in the type of cell divisions observed in progenitors upon infection as we no longer saw divisions along the z-axis (or dorsal-ventral axis; *Figure 4G*).

A custom quantitative image analysis algorithm was developed and used to explore spatial differences in the location and orientation of cell division between progenitors in the LGs from wild-type control and infected larvae (*Figure 4—figure supplement 2*; see Materials and methods). First, the LG was segmented into regions based on distance and location relative to the heart tube and

the posterior end of the LG. Second, the number of progenitor cell divisions in each segment was determined for multiple LGs from wild-type control and infected larvae (*Figure 4J*; see Methods). This analysis showed a uniform reduction in progenitor cell divisions throughout the LGs following infection (*Figure 4J*; correlation coefficient for changes in distribution upon infection compared to control = 0.44 consistent with weak correlation; see Methods). Analysis of the distribution of division frequency of progenitors and their relative angle to the heart tube showed that reduced progenitor divisions following infection were uniform across all angles within the LGs and there was no change in the duration of mitotic events at any division angle relative to the heart tube (*Figure 4K–M*). The orientation of these divisions remained biased following infection (*Figure 3—figure supplement 1G*). Taken together, the data show that, when compared to uninfected controls, following infection: (1) There is a reduction in the number of progenitors undergoing division. (2) The division of progenitors is more likely to occur along the plane formed by the anterior-posterior and right-left axes. (3) The duration and orientation of mitotic events in progenitors are unchanged. Importantly, these results show that changes in proliferation are unlikely to account for the increased number of mature blood cells produced following infection. Consequently, changes in differentiation are likely the main driver for the increased mature blood cell production following infection.

## Quantitative imaging identifies two types of blood cell differentiation in the LG

We analyzed mature blood cell differentiation in LGs from wild-type larvae in real time using quantitative imaging of genetically encoded fluorescent reporters for cell identity and signaling activity (*Figure 2—figure supplement 1B–C*; see Methods). To first confirm that the ex vivo culture condition did not impact differentiation, we compared differentiation under ex vivo and physiological in vivo conditions over the course of 12 hr and we observed comparable levels of differentiation (*Figure 1—figure supplement 3G*). Next, the expression levels of the JAK-STAT reporter and progenitor marker dome-MESO-GFP as well as the early blood cell differentiation marker eater-dsRed were simultaneously tracked in individual cells for 12–14 hr. This allows us to observe individual differentiation events, where the expression of the progenitor marker dome-MESO-GFP declined in a cell, while the expression level of eater-dsRed increased (*Figure 5A*; *Video 5*). A plot of the expression levels of dome-MESO-GFP (plotted on the y-axis) and eater-dsRed (plotted on the x-axis) at each time point collected from an individual cell revealed the differentiation trajectory from progenitor to differentiated cell (*Figure 5B*). Simultaneous tracking of dome-MESO-GFP and eater-dsRed in differentiating cells identified two types of cell fate trajectories, which we refer to as sigmoid and linear (*Figure 5C–F*; see Materials and methods). In a cell following a sigmoid trajectory, named after the shape of a sigmoid function curve (*Figure 5D*), the initial level of dome-MESO-GFP is high while the levels of eater-dsRed is low (*Figure 5C*; *Video 6*). As time passes in a cell following a sigmoid differentiation trajectory, the level of dome-MESO-GFP decreases and after a short delay the level of eater-dsRed increases (*Figure 5C*). A key feature of cells undergoing a sigmoid differentiation trajectory is that the differentiation process is broken down into an initial slow phase and then a rapid fast phase (*Figure 5D*). This is best visualised by

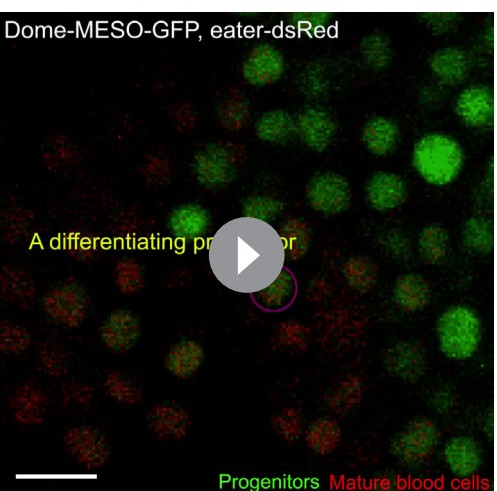

**Video 5.** Long-term tracking of blood progenitor differentiation in a wild-type LG. Representative video of a differentiating blood progenitor (the cell was green at the beginning) turning into a differentiated mature blood cell (the cell became red in the end) in a live intact LG. Blood progenitors were marked by dome-MESO-GFP (green). Mature hemocytes were marked by eater-dsRed (red). The tracked progenitor was highlighted using a pink ROI by TrackMate throughout the recording. The LG was obtained from an early 3rd instar larva (of genotype *dome-MESO-GFP, eater-dsRed*) raised at 25 °C, dissected, immediately mounted and imaged. Scale Bar: 10 µm.

https://elifesciences.org/articles/84085/figures#video5

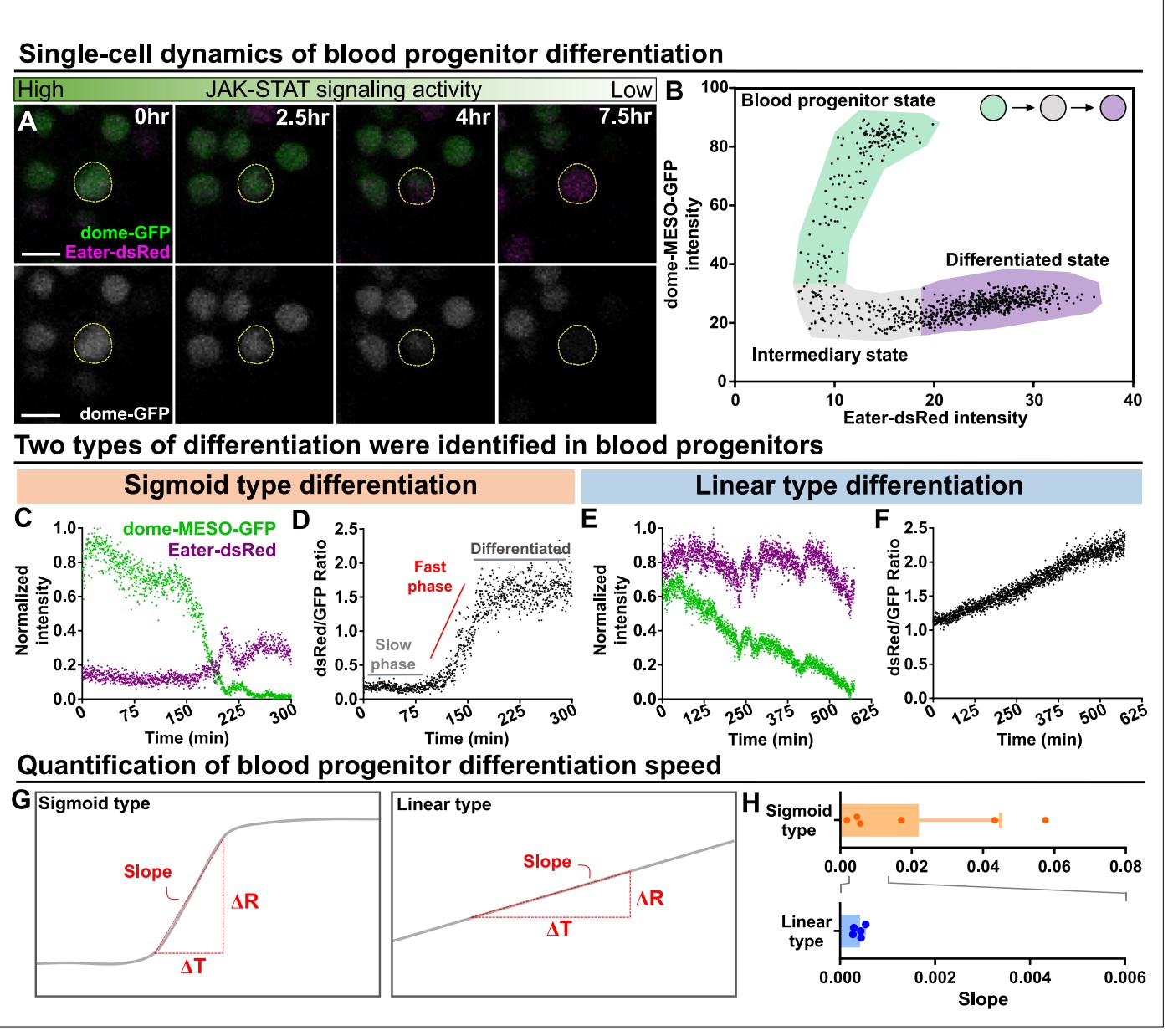

**Figure 5.** Long-term imaging identifies distinct dynamics of blood progenitor differentiation. (**A**) Time-lapse images of blood progenitor differentiation over the course of 7.5 hrs. Blood progenitors labelled with dome-MESO-GFP (green in top panel, white in below panel). Mature hemocytes labelled with eater-dsRed (magenta). (**B**) Example showing real-time tracking of dome-MESO-GFP and eater-dsRed fluorescent intensities of a progenitor over time. Each dot represents a single time point. (**C–D**) Examples showing normalized intensities of dome-MESO-GFP and eater-dsRed (**C**) and dsRed/GFP ratio (**D**) in blood progenitors undergoing sigmoid type differentiation over the course of roughly 300 min. Each dot represents the fluorescent intensity (**C**) or ratio (**D**) of dome-MESO-GFP or eater-dsRed at that time point. (**E–F**) Examples showing normalized intensities of dome-MESO-GFP and eater-dsRed (**E**) and dsRed/GFP ratio (**F**) in blood progenitors undergoing linear type differentiation over the course of roughly 560 min. Each dot represents the fluorescent intensity (**E**) or ratio (**F**) of dome-MESO-GFP or eater-dsRed at that time point. (**G**) Schematic illustrating the method used to quantify the rate of differentiation in both types (left panel: sigmoid type, right panel: linear type). Changes of dsRed/GFP ratio (ΔR) over a period of time (ΔT) were used to calculate the slope (see Materials and methods). (**H**) Quantification of differentiation rate (or slope) of the sigmoid type (n=6 blood progenitors collected from 8 videos) and the linear type differentiation (n=5 blood progenitors collected from 8 videos) in wild-type LGs. Genotype of the LG was *dome-MESO-GFP; eater-dsRed*. Error bars indicate S.D from the mean. See also *Video 6* and *Video 7*.

The online version of this article includes the following source data and figure supplement(s) for figure 5:

**Source data 1.** Raw data of *Figure 5B, C, D, E, F and H*.

**Figure supplement 1.** Linear and sigmoid type differentiation can occur in parallel or at distinct time points.

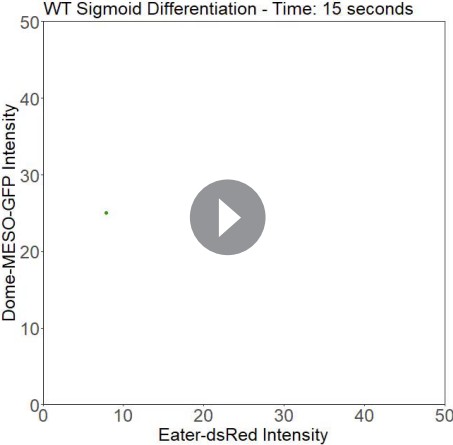

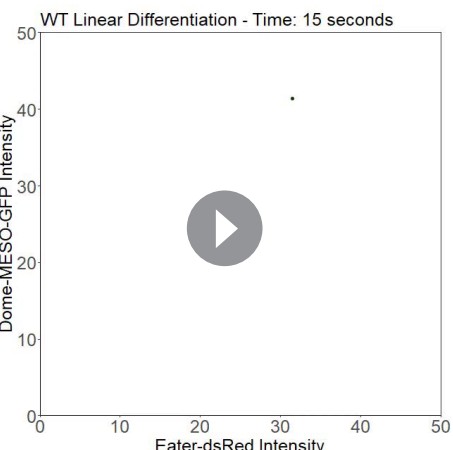

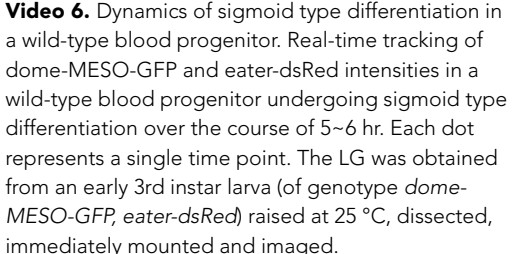

**Video 6.** Dynamics of sigmoid type differentiation in a wild-type blood progenitor. Real-time tracking of dome-MESO-GFP and eater-dsRed intensities in a wild-type blood progenitor undergoing sigmoid type differentiation over the course of 5~6 hr. Each dot represents a single time point. The LG was obtained from an early 3rd instar larva (of genotype *dome-MESO-GFP, eater-dsRed*) raised at 25 °C, dissected, immediately mounted and imaged.

https://elifesciences.org/articles/84085/figures#video6

**Video 7.** Dynamics of linear type differentiation in a wild-type blood progenitor. Real-time tracking of dome-MESO-GFP and eater-dsRed intensities in a wild-type blood progenitor undergoing linear type differentiation over the course of 7~8 hr. Each dot represents a single time point. The LG was obtained from an early 3rd instar larva (of genotype *dome-MESO-GFP, eater-dsRed*) raised at 25 °C, dissected, immediately mounted and imaged.

https://elifesciences.org/articles/84085/figures#video7

calculating the ratio of dsRed to GFP in the cell as a function of time. The graph shows the characteristics of sigmoid function shape from which the name of this type of differentiation trajectory is derived. Importantly, the sigmoid differentiation trajectory results in a rapid shift from a high dome-MESO-GFP and low eater-dsRed cell to a differentiated low dome-MESO-GFP and high eater-dsRed cell (*Figure 5C–D*).

In comparison, in a cell following a linear differentiation trajectory, the relative levels of both dome-MESO-GFP and eater-dsRed are high to begin with (*Figure 5E*; *Video 7*). As time passes in a cell following a linear differentiation trajectory, the level of dome-MESO-GFP decreases while the level of eater-dsRed remains high. A key feature of cells undergoing a linear differentiation trajectory is that the differentiation process exhibits a uniform rate (*Figure 5F*). This is best visualised by calculating the ratio of dsRed to GFP in the cell as a function of time. The graph shows the characteristics of linear function shape from which the name of this differentiation trajectory is derived (*Figure 5F*). Importantly, the linear trajectory results in a gradual shift from a high dome-MESO-GFP and high eater-dsRed cell to a differentiated low dome-MESO-GFP and high eater-dsRed cell (*Figure 5E and F*). The rate of differentiation from a progenitor to a blood cell can be quantified by calculating the slope from the graph of the ratio of dsRed to GFP in the cell as a function of time during the phase where differentiation occurs (*Figure 5G*). This analysis shows that the sigmoid trajectory differentiation occurs at a rapid rate over a short time frame, while the linear trajectory differentiation is slower and more gradual (*Figure 5H*).

Importantly, the two differentiation trajectories appeared distinct and cells undergoing linear differentiation are not simply in the later phase of sigmoid type differentiation where eater-dsRed is high, but dome-MESO-GFP is already low. This is evident because there are features that distinguish the two trajectories in the later phases. Specifically: (1) A key characteristic of linear differentiation trajectory is that the dome-MESO-GFP declines throughout the process. In contrast, in the later phases of the sigmoid differentiation trajectory, dome-MESO-GFP levels become stable (comparing *Figure 5C–E*). The kinetics of linear type differentiation is therefore different from the later phases of sigmoid type differentiation. (2) From the middle to late phase of the sigmoid differentiation trajectory, eater-dsRed levels go up after dome-MESO-GFP levels are already low or still decreasing. In contrast, in the linear trajectory eater-dsRed levels are stable in the later parts of the trajectory.

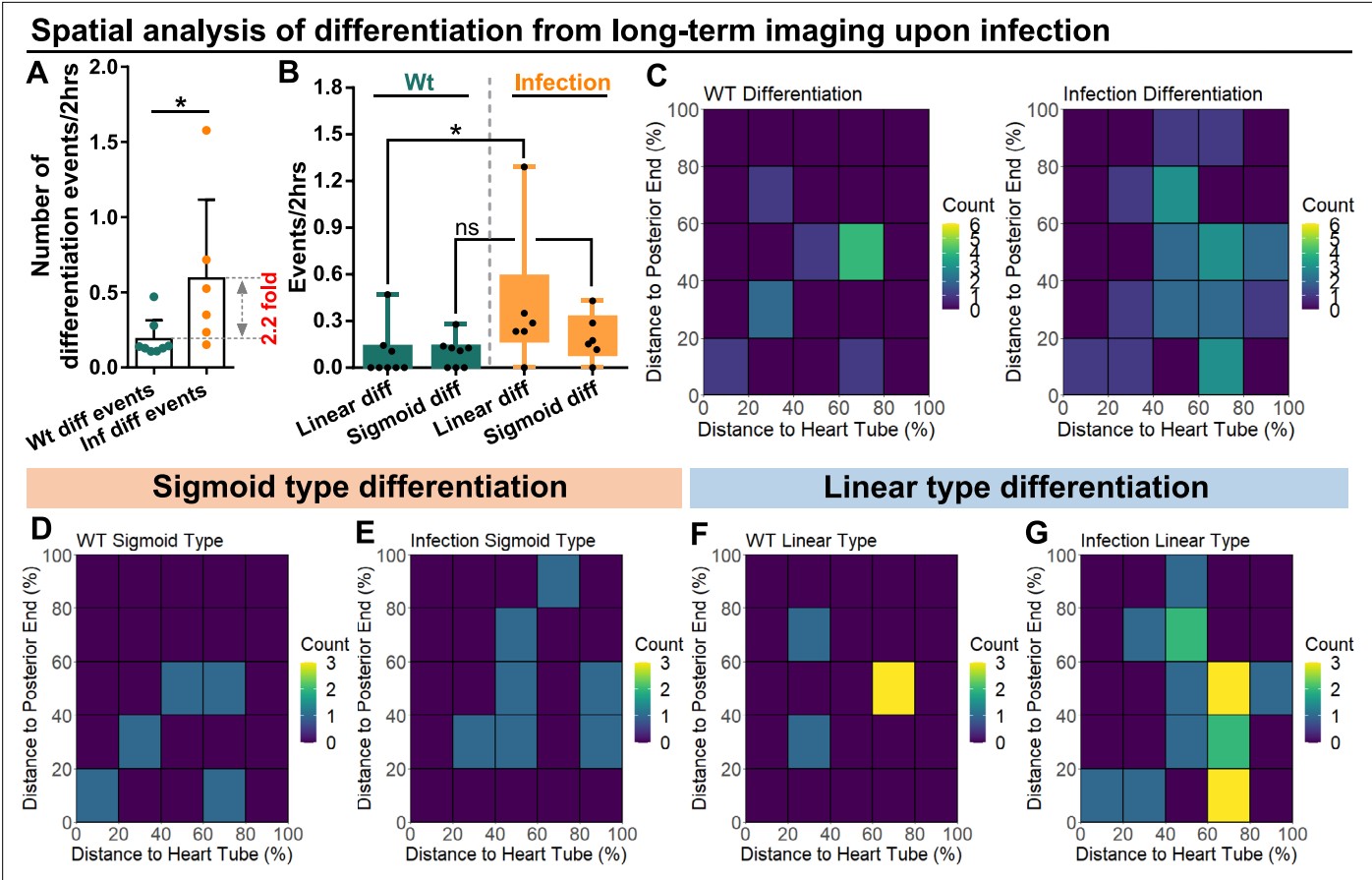

**Figure 6.** Spatial distribution of sigmoid and linear type differentiations upon immune activation. (**A**) Quantification of the number of blood progenitor differentiation events recorded in LGs from wild-type control (n=8 videos that capture 11 differentiation events in total) and *E. coli* infected larvae (n=6 videos that capture 25 differentiation events in total). p-Value = 0.0117, determined using Mann-Whitney-Wilcoxon test. (**B**) Quantification of linear and sigmoid differentiations events in LGs from wild-type control (n=8 videos) and *E. coli* infected larvae (n=6 videos). p-Values = 0.0426 and 0.1592, determined using Kolmogorov Smirnov test. (**C–G**) Heat maps collating data from long-term videos that show the number of total differentiation events (**C**), sigmoid type differentiation events (**D, E**), and linear type differentiation events (**F, G**) recorded in specific regions of LGs from wild-type control (C left panel, **D, F**) and *E. coli* infected larvae (C right panel, **E, G**). Genotype of the LG was *dome-MESO-GFP; eater-dsRed*. * indicates p<0.05. ns indicates non-significant, p>0.05.

The online version of this article includes the following source data and figure supplement(s) for figure 6:

**Source data 1.** Raw data of *Figure 6A and B*.

**Figure supplement 1.** Ex vivo cultured LGs demonstrate comparable differentiation trends to in vivo LGs following infection.

**Figure supplement 1—source data 1.** Raw data of *Figure 6—figure supplement 1*.

Moreover, spatiotemporal analysis of differentiation trajectories suggests they occur separately, that is not in a consecutive manner whereby cells undergo a sigmoid trajectory first and subsequently a linear trajectory. In particular, in the LG regions where we identified cells following different types of trajectories, tracking the trajectories of individual cells shows that 1. They exhibit a single distinct type of differentiation throughout (*Figure 5—figure supplement 1A*) and 2. Cells undergo sigmoid or linear type differentiation either in parallel (see Cell1 and Cell2 in 1 of Wt in *Figure 5—figure supplement 1A*) or at different time points (see Box2 of Wt in *Figure 5—figure supplement 1A*). Taken together, these results identify two distinct types of differentiation events in the LG.

## Infection changes cell differentiation patterns in the LG

Live imaging experiments were used to track and quantify differentiation events in LGs from wild-type control and infected larvae. First, we confirmed that the general differentiation trends following infection were similar between LGs in the ex vivo culture system and physiological in vivo conditions

(*Figure 6—figure supplement 1*). Second, we noted a significant (>2-fold) increase in the number of differentiation events observed in LGs following infection (*Figure 6A*). Third, we determined the spatial distribution of differentiation events in the LGs from wild-type control and infected larvae. A general increase was seen in differentiation events, especially near the area that would correspond to the MZ-CZ boundary (around 60% distance to heart tube; *Figure 6B–C*; correlation coefficient for changes in distribution upon infection compared to control = 0.49 consistent with weak correlation; see Materials and methods). Finally, the spatial distribution of sigmoid and linear trajectory differentiation events was analysed separately (*Figure 6D–G*). This revealed that the spatial distribution of sigmoid trajectory differentiation events was not greatly altered by infection (*Figure 6D–E*; correlation coefficient between heat maps of 6D and 6E=0.14 consistent with no correlation). In comparison, the spatial distribution of linear trajectory differentiation events showed differences following infection (*Figure 6F–G*; correlation coefficient between heat maps of 6 F and 6G=0.45 consistent with weak correlation). In particular, there was an increase in the frequency of differentiation events, especially near the area that would correspond to the MZ-CZ boundary. In addition, spatiotemporal analysis of the two types of differentiation trajectories following infection showed that they took place either in parallel (for example Cell 2 and Cell3 in Box 1 of Infection group in *Figure 5—figure supplement 1B*) or at different time points (*Figure 5—figure supplement 1B*).

Next, we tracked the differentiation trajectory of individual cells by measuring in real time the expression levels of dome-MESO-GFP and eater-dsRed in wild-type control and infected larvae (*Figure 7A–D*; *Figure 7—figure supplement 1A–D*; *Figure 7—figure supplement 2A–D* for Wt and 2E-H for infection group; *Videos 6–9*). As in control LGs, both the sigmoid and linear differentiation trajectories were observed in infected LGs. However, the differentiation trajectories exhibited some variance in infected versus control larvae. For example, progenitor cells following the sigmoid trajectory in infected LGs exhibited a prolonged intermediary phase, during which both dome-MESO-GFP and eater-dsRed were expressed at low levels (*Figure 7A–A″ , and B–B″*, quantified in *Figure 7E*; *Figure 7—figure supplement 1A and C*; *Videos 6 and 8*). Moreover, as a result of the prolonged intermediary phase, the average rate of differentiation for the sigmoid trajectory was lower upon infection (*Figure 7F*). We also observed a slightly modified linear type trajectory in LGs from infected larvae compared to controls. In particular, while in controls the expression of eater-dsRed was relatively constant but that of dome-MESO-GFP declined with time (*Figure 7C–C″*; *Figure 7—figure supplement 1B*; *Figure 7—figure supplement 2A–B*; *Video 7*), in infected LGs the expression of eater-dsRed went up at first and dome-MESO-GFP expression declined later (*Figure 7D–D″*; *Figure 7—figure supplement 1D*; *Figure 7—figure supplement 2E–F*; *Video 9*). This does not modify the overall rate of differentiation (*Figure 7G*) but does result in a~50% increase in the ratio of expression of eater-dsRed to dome-MESO-GFP (*Figure 7C″ , and D″*; *Figure 7—figure supplement 2A and B, 2E and 2F*). Importantly, upon infection, there is around 20% increase in the proportion of differentiation events that follow the linear trajectory and a corresponding decrease in the number of differentiation events that follow the sigmoid trajectory (*Figure 7H*). Taken together, the data is consistent with a model whereby infection causes higher differentiation in the LGs not by increasing the rate of differentiation but rather by inducing a shift from one type of differentiation, the sigmoid trajectory, to another, the linear trajectory.

To understand why there was a reduction in the number of sigmoid trajectory differentiation events, we applied a modified version of a technique known as histo-cytometry which presents in vivo derived data in a similar data format from flow cytometry (see Materials and methods; *Stoltzfus et al., 2020*). We imaged LGs expressing eater-dsRed and dome-MESO-GFP and performed automated image analysis to determine the relative amounts of these markers in individual cells in the LG (*Figure 7I–L*; in total 2500 cells captured from 6 primary lobes of LGs in both wt and infection groups; see Materials and methods). When compared to the wild-type control, the relative distribution of expression profiles of eater-dsRed and dome-MESO-GFP was greatly altered by infection. Specifically, there was an overall increase in the expression of eater-dsRed following infection in many cells in the LGs (*Figure 7K–L*). This shift to higher eater-dsRed can indicate immune activation, as *eater* is transcriptionally activated as part of the immune response following infection (*Kocks et al., 2005*; *Kroeger et al., 2012*; *Ye and McGraw, 2011*), but the shift is also consistent with a greater proportion of progenitors undergoing differentiation. Cells were classified into four general categories based on their differentiation profile: GFP$^{HIGH}$dsRed$^{LOW}$ (most stem cell-like), GFP$^{LOW}$dsRed$^{HIGH}$ (most differentiated), GFP$^{HIGH}$dsRed$^{HIGH}$ and

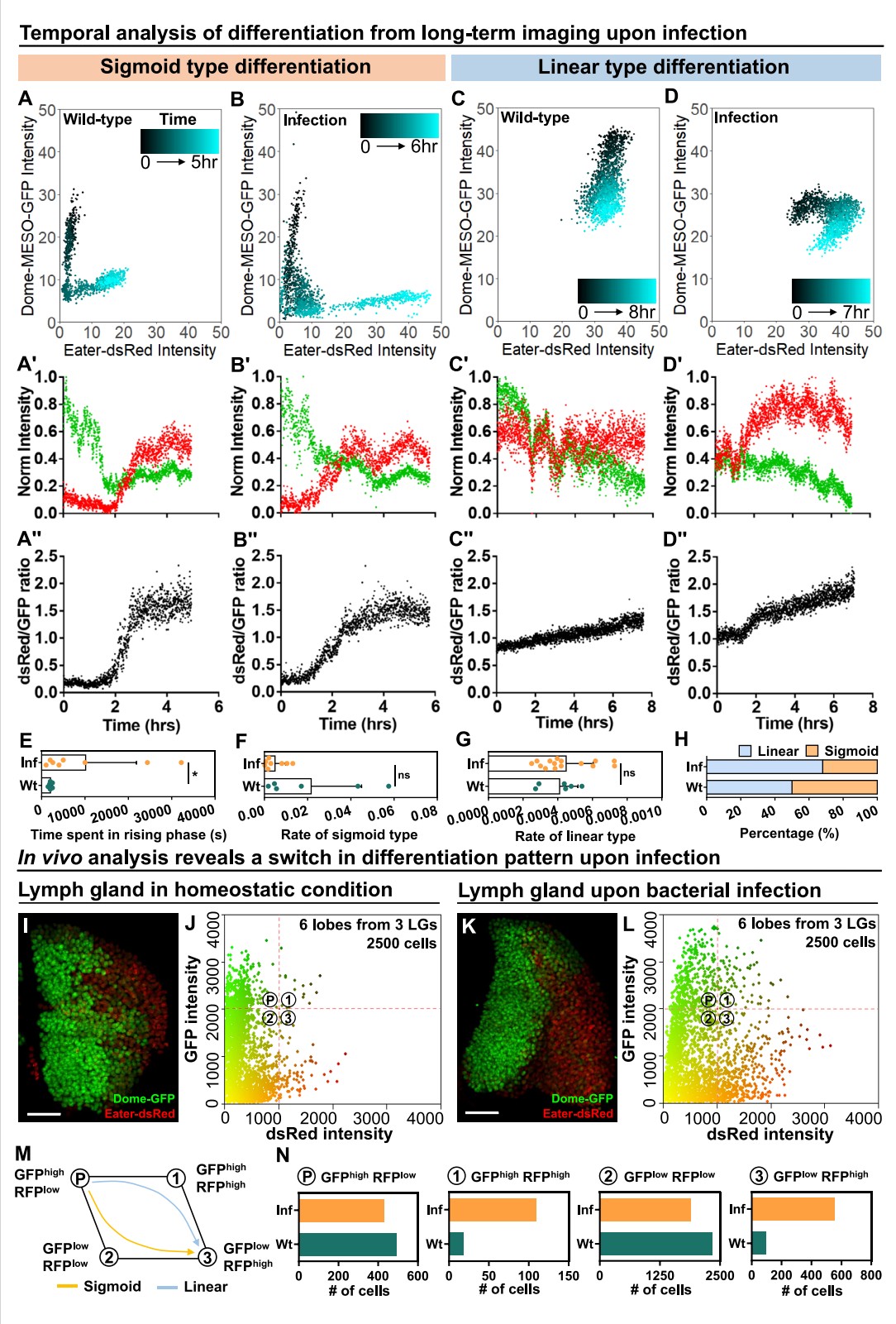

**Figure 7.** Temporal control of sigmoid and linear type differentiations upon bacterial infection. (**A–B**) Representative sigmoid differentiation trajectories in blood progenitors from LGs of wild-type control (**A-A‴**) or *E. coli* infected larvae (**B-B‴**) over the course of 5–6 hr. (**C–D**) Representative linear differentiation trajectories in blood progenitors from LGs of wild-type control (**C-C″**) and *E. coli* infected larvae (**D-D″**) over the course of 7–8 hr. Dome-MESO-GFP and eater-dsRed fluorescent intensities are used to visualize differentiation kinetics. Each dot represents a single time point.

*Figure 7 continued on next page*

*Figure 7 continued*

Blood progenitors are labelled with dome-MESO-GFP. Mature hemocytes are labelled with eater-dsRed. (**E–G**) Quantification of the duration of the fast differentiation phase in progenitors undergoing the sigmoid differentiation trajectory (E, n=6 and 8 progenitors from LGs of wild-type control and *E. coli* infected larvae; p-value = 0.0226), the differentiation rate measured in progenitors undergoing sigmoid type differentiation trajectory (F, n=6 and 8 progenitors from LGs of wild-type control and *E. coli* infected larvae, respectively; p-value = 0.2731), and the differentiation rate measured in progenitors undergoing a linear type differentiation trajectory (G, n=6 and 15 progenitors from LGs of wild-type control and *E. coli* infected larvae, respectively; p-value = 0.6613). (**H**) Quantification of the percentage of sigmoid or linear type differentiation trajectories observed in LGs from wild-type control and *E. coli* infected larvae. (**I–J**) Representative image (**I**) and scatterplot (**J**) of wild-type control LGs (n=2500 cells in total analyzed from 6 primary lobes of 3 LGs, see Materials and methods). (**K–L**) Representative image (**K**) and scatterplot (**L**) of LGs from *E. coli* infected larvae (n=2500 cells in total analyzed from 6 primary lobes of 3 LGs, see Materials and methods). (**M**) Schematic illustrating the two observed differentiation trajectories (sigmoid and linear). Based on their fluorescent intensities of GFP (progenitor fate marker) and dsRed (differentiated state marker), cells in the LG are categorized into 4 groups: GFP$^{high}$ RFP$^{low}$, GFP$^{high}$ RFP$^{high}$, GFP$^{low}$ RFP$^{high}$, GFP$^{low}$ RFP$^{low}$. P: blood progenitors. (**N**) Quantification of the total number of cells in each quadrant of (**J**) and (**L**) from the LGs of wild-type control and *E. coli* infected larvae. p Values in (**E–G**) were determined using Kolmogorov Smirnov test. * indicates p<0.05. ns indicates non-significant, p>0.05. Scale bars in (**I**) and (**K**) represent 50 μm. Error bars indicate S.D from the mean. Genotype of the LG was *dome-MESO-GFP; eater-dsRed*. See also *Videos 6–9*.

The online version of this article includes the following source data and figure supplement(s) for figure 7:

**Source data 1.** Raw data of *Figure 7A, A', A", B, B', B", C, C', C", D, D', D", E, F, G, H, J, L and N*.

**Figure supplement 1.** Tracking differentiation events at single cell resolution in real time following infection.

**Figure supplement 2.** The normalization method preserves the original trend of cell fate markers during differentiation in wild-type condition and upon infection.

**Figure supplement 2—source data 1.** Raw data of *Figure 7—figure supplement 1A, B, C, D, E, F, G, H*.

GFP$^{LOW}$dsRed$^{LOW}$ (both intermediate stages; *Figure 7M*). While a sigmoid differentiation trajectory proceeds as GFP$^{HIGH}$dsRed$^{LOW}$ to GFP$^{LOW}$dsRed$^{LOW}$ to GFP$^{LOW}$dsRed$^{HIGH}$, the linear trajectory proceeds as GFP$^{HIGH}$dsRed$^{LOW}$ to GFP$^{HIGH}$dsRed$^{HIGH}$ to GFP$^{LOW}$dsRed$^{HIGH}$ (*Figure 7M*). Notably, following infection, the relative overall population of GFP$^{HIGH}$dsRed$^{HIGH}$ increased (18 cells in wt LGs and 109 cells in LGs following infection), while GFP$^{LOW}$dsRed$^{LOW}$ decreased slightly (2337 cells in wt LGs and 1888 cells in LGs following infection) and there was a rise in the number of differentiated GFP$^{LOW}$dsRed$^{HIGH}$

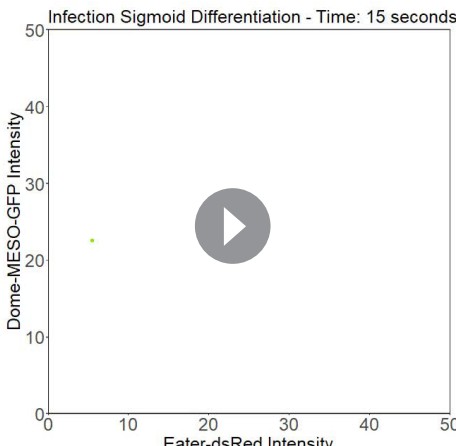

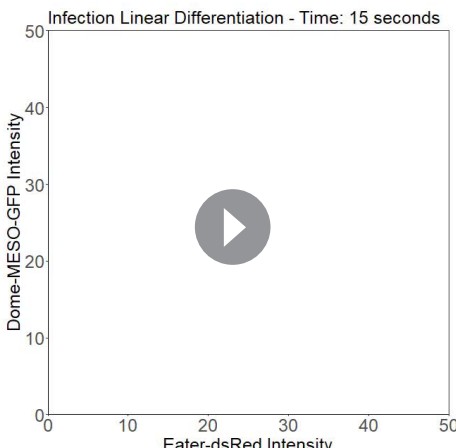

**Video 8.** Dynamics of sigmoid type differentiation in a blood progenitor upon *E. coli* infection. Real-time tracking of dome-MESO-GFP and eater-dsRed intensities in a blood progenitor undergoing sigmoid type differentiation in the LG derived from an *E. coli* infected larva over the course of 5~6 hr. Each dot represents a single time point. The LG was obtained from an early 3rd instar larva (of genotype *dome-MESO-GFP, eater-dsRed*) raised at 25 °C, dissected, immediately mounted and imaged.

https://elifesciences.org/articles/84085/figures#video8

**Video 9.** Dynamics of linear type differentiation in a blood progenitor upon *E. coli* infection. Real-time tracking of dome-MESO-GFP and eater-dsRed intensity in a blood progenitor undergoing linear type differentiation in the LG derived from an *E. coli* infected larva over the course of 7~8 hr. Each dot represents a single time point. The LG was obtained from an early 3rd instar larva (of genotype *dome-MESO-GFP, eater-dsRed*) raised at 25 °C, dissected, immediately mounted and imaged.

https://elifesciences.org/articles/84085/figures#video9

cells (97 cells in wt LGs and 557 cells in LGs following infection; *Figure 7N*: data summarized from each quadrant of 7 J and 7 L; see Materials and methods). Overall, these observations suggest a possible link between the two main features seen upon infection, increased overall differentiation and an increased proportion of cells undergoing the linear trajectory of differentiation. In particular, these findings confirm the view that there are different subpopulations of progenitors in the LG (*Cho et al., 2020*; *Blanco-Obregon et al., 2020*), and raise the possibility that changes in the proportion of these different subpopulations are involved in the activation of mature blood cell differentiation following infection.

## Discussion

By employing organ culture, using genetically encoded markers for cell cycle, proliferation and differentiation, and implementing quantitative image analysis, we are able to study the process of fly hematopoiesis at a single cell resolution in its endogenous context. This has allowed us to observe features of hematopoiesis that only become apparent upon system- and organ-level analysis. Our observations lead us to four main conclusions. First, our results illustrate that certain populations of blood progenitors in the fly can undergo symmetric cell divisions. Second, we find that the timing of blood progenitor division is more likely to occur once cells reach a specific cell size and the division is spatially oriented with respect to the heart tube and anatomical axes. Third, we identify and characterise two distinct modes of differentiation. Fourth, we show that the induction of mature blood cell production in response to infection is not achieved by modulating progenitor proliferation or speed of differentiation but by increasing the size of a population of progenitors that expressed high levels of both differentiated and progenitor cell markers.

The ex vivo culture and imaging protocol we describe provides a powerful new way to study hematopoiesis over a prolonged time using quantitative imaging. Although such an approach holds much promise, its use calls for caution and for the consideration of certain caveats. For example, following infection, while LGs in vivo would be exposed to systemic immune signals, cultured LGs may not have access to such continuous extrinsic signals which may change their behaviour. During our work, we have looked at a number of parameters of differentiation, proliferation, and tissue homeostasis and did not see any strong differences between in vivo and ex vivo organ culture. Nonetheless such differences may exist in some circumstances. Another caveat is that due to the combination of tagged markers available for live imaging, we were only able to follow differentiation events leading to plasmatocytes and not crystal cell fate. Future work will focus on addressing this to provide a more complete view of blood cell differentiation. Finally, our observations show the existence of a large population of LG cells that expresses low levels of both dome-MESO-GFP and eater-dsRed. This would suggest that a substantial proportion of progenitors begin to differentiate but pause at this stage. However, we did not observe in our live trajectory tracking experiments this sort of paused trajectory. We can only speculate at this point that cells in the paused trajectory are produced before the early third instar larval stage, which is the stage we chose for live imaging. In our future work, we will extend our analysis to earlier larval stages, which we hope will test this hypothesis directly.

Our work supports earlier studies that described the presence of distinct subpopulations of progenitors in the LG (*Cho et al., 2020*; *Blanco-Obregon et al., 2020*). According to this emerging model, the progenitor pool is not homogenous during hematopoiesis but rather contains subpopulations at different levels of differentiation. Importantly, these studies used single-cell transcriptomic, a different approach to ours, but also suggest that multiple paths exist for blood progenitors to differentiate into plasmatocytes (*Cho et al., 2020*). Specifically, they identified a path that contains an intermediate mixed lineage stage of differentiation and a more direct path that does not include intermediate steps (*Cho et al., 2020*). Additional subpopulations that have been proposed to exist in the LG include the intermediate progenitors (*Krzemien et al., 2010*; *Sinenko et al., 2009*; *Sharma et al., 2019*; *Cho et al., 2020*; *Girard et al., 2021*; *Spratford et al., 2021*) as well as the distal progenitors (*Blanco-Obregon et al., 2020*), which express a mixture of both progenitor and differentiated cell markers. Intermediate progenitors express the progenitor marker Dome and early differentiation marker Pxn but not mature blood cells markers like P1 or Lz (*Sharma et al., 2019*; *Sinenko et al., 2009*; *Krzemien et al., 2010*). Distal progenitors also exhibit a mixed fate: expressing the progenitor marker Dome, but also hallmarks of differentiated cells such as the expression of the plasmatocyte marker eater and absence of the progenitor marker Tep4 (*Blanco-Obregon et al., 2020*). Other studies suggested the

existence of PSC-dependent and PSC-independent progenitors (*Baldeosingh et al., 2018*; *Mandal et al., 2007*). In our live imaging experiments, we observed a substantial population of cells in the LG that simultaneously express high levels of both progenitor and differentiated cell markers, which likely includes cells belonging to one or more of these mixed lineage subpopulations (*Cho et al., 2020*; *Blanco-Obregon et al., 2020*). Our studies suggest that these mixed lineage cells play a crucial role in hematopoiesis as they represent the linear differentiation trajectory which drives increased differentiation in response to infection. Consistent with this idea, spatial analysis of where progenitors that follow the different trajectories are located shows that the linear trajectory occurs mostly in the region thought to hold intermediate progenitors.

Surprisingly, the linear differentiation trajectory is substantially slower than the sigmoid differentiation trajectory. This appears to be in contradiction to another one of our observations, that a very large proportion of the progenitors following the sigmoid trajectory are found in an intermediate state where both the progenitor and differentiation markers are expressed at low levels (see *Figure 7J* area #2). The accumulation of sigmoid-trajectory progenitors at the intermediate phase would suggest this is a long-lasting phase, but this is not what we saw in our direct tracking of differentiation trajectories. We propose that a possible explanation to resolve this potential contradiction is that only a small proportion of the cells that are found in the intermediate state where both the progenitor and differentiation markers are expressed at low levels go on to differentiate. Furthermore, it is unclear why the linear, slower trajectory, would be favored under conditions where we would expect a need for rapid production of immune cells. We can speculate the benefits of expanding the population of cells undergoing the linear trajectory exceed the disadvantages conferred by the slower differentiation time. We would propose, based on our observations, that the various subpopulations with mixed progenitor and differentiated cell fate have an important role during infection by acting as transit amplifying cells that allow rapid induction of the cellular immune response. Understanding the behavior of these intermediate state cells should be a focus of future investigation.

Multiple systems and approaches have been used to track HSCs and blood progenitors during hematopoiesis in their native environment in real time. Key examples include studies in zebrafish that used intravital imaging (*Zhang and Liu, 2011*; *Frame et al., 2017*), studies in mice that combined diverse approaches such as inducible lineage tracing, flow cytometry, and single-cell RNA sequencing (*Upadhaya et al., 2018*), as well as mouse studies based on intravital imaging of the bone marrow (*Christodoulou et al., 2020*). Zebrafish have proven to be particularly useful for live studies of hematopoiesis, due to the relative ease of intravital imaging, and its wealth of transgenic tools (*Zhang and Liu, 2011*). Zebrafish have been a powerful system for studying the embryonic development of HSCs and the hematopoietic niche as well as for drug, chemical and genetic screening (*Arulmozhivarman et al., 2016*). In addition, zebrafish have been proven to be very useful for modeling blood malignances and tracking their development and disease progression (*Robertson et al., 2016*; *Gore et al., 2018*). In the mouse system, which holds many challenges for intravital imaging, alternative approaches have been used to capture the dynamics of the process of hematopoiesis (*Upadhaya et al., 2018*; *Grinenko et al., 2018*). For example, *Upadhaya et al., 2018* used a drug inducible HSC labeling technique to isolate HSCs and their progeny at set time points and follow the transcriptional landscape of the progenitors as they progress along their developmental trajectory (*Upadhaya et al., 2018*). This type of analysis yields several intriguing insights into hematopoiesis, such as the differences in the time it takes for various blood lineages to differentiate (*Upadhaya et al., 2018*). Moreover, despite the technical challenges, there have been several successful attempts to image the process of hematopoiesis in vivo by using the bone marrow of the mouse skull. While initially this approach was limited to the use of isolated, labelled, and transplanted HSCs (*Lo Celso et al., 2009*), more recent studies used an endogenously labelled HSC line (*Christodoulou et al., 2020*). Although they constitute important technical breakthroughs, these studies still suffer from several challenges and allow the visualization of a relatively short time window compared to the actual time it takes progenitors to differentiate in the mouse. Consequently, these studies were limited to describing the architecture of the bone marrow niche and the location of HSCs within it (*Lo Celso et al., 2009*), or to general descriptions of a small subset of cell behaviors such as HSC/progenitor motility and expansion (*Christodoulou et al., 2020*). Our approach offers the ability to perform real time functional studies that can complement observations from these other models of hematopoiesis.

In particular, compared with these earlier studies our approach offers several key innovations. First is our ability to track multiple markers simultaneously in a quantitative way during long-term live imaging. Specifically, our approach allows us to quantitatively track, for 12 or more hours, markers of cell fate in combination with multiple other markers for proliferation, metabolism, cell signaling, and cell morphology. Moreover, the relatively short duration of the differentiation process in the fly, approximately 6–8 hr versus 1–3 weeks for various leukocyte lineages in the mouse (*Upadhaya et al., 2018*), allows us to observe differentiation in its entirety. Second, the ability to track a large number of progenitors and quantitate both their behavior and the expression of markers using imaging analysis tools allows the deployment of system-level approaches. This offers the capability to track hematopoiesis at a cellular and even subcellular spatial resolution and a temporal resolution of a few seconds, well beyond previous studies. Third, the ability to combine these powerful analysis tools with an infection model facilitates the ability to visualize the induction of the cellular branch of the immune response in real time in order to elucidate the underlying mechanisms. Fourth, the vast genetic toolkit and short generation time of the fly, the accessibility of the LG multi-organ co-culture system to drug (*Ho et al., 2021*) and organ-organ communication studies, and the detailed and extensive transcriptomic analysis of blood cell differentiation (*Cho et al., 2020*; *Girard et al., 2021*) all make it a superb system for real time analysis of hematopoiesis. Specifically, a major goal of our future work will focus on combining the analysis pipeline we describe here with markers and tools to analyze and manipulate the various cell signaling pathways that have been implicated in the regulation of hematopoiesis under homeostatic, infection, and pathogenic conditions.

# Materials and methods

## Key resources table

| Reagent type (species) or resource | Designation | Source or reference | Identifiers | Additional information |
|---|---|---|---|---|
| Genetic reagent (*Drosophila melanogaster*) | Tep4-Gal4 | *Avet-Rochex et al., 2010* | Flybase ID: FBti0037434 | Gift from Dr. Lucas Waltzer, Université Clermont Auvergne, France |
| Genetic reagent (*Drosophila melanogaster*) | dome-MESO-Gal4 | *Hombría et al., 2005* | Flybase ID: FBtp0146166 | Gift from Dr. Lucas Waltzer, Université Clermont Auvergne, France |
| Genetic reagent (*Drosophila melanogaster*) | eater-dsRed | *Kroeger et al., 2012 Tokusumi et al., 2009* | Flybase ID: FBtp0084524 | Gift from Dr. Elio Sucena, Instituto Gulbenkian de Ciência, Portugal |
| Genetic reagent (*Drosophila melanogaster*) | dome-MESO-GFP.nls | *Oyallon et al., 2016* | Flybase ID: FBtp0142446 | Gift from Dr. Michele Crozatier, Université de Toulouse, France |
| Genetic reagent (*Drosophila melanogaster*) | gstD-GFP | *Sykiotis and Bohmann, 2008* | Flybase ID: FBtp0069371 | Gift from Dr. Dirk Bohmann, University of Rochester Medical Center, USA |
| Genetic reagent (*Drosophila melanogaster*) | dome-MESO-LacZ | *Hombría et al., 2005* | Flybase ID: FBtp0022619 | Gift from Dr. Nancy Fossett, University of Maryland, Baltimore, USA |
| Genetic reagent (*Drosophila melanogaster*) | HmlΔ-dsRed.nls | *Makhijani et al., 2011* | Flybase ID: FBtp0150011 | Gift from Dr. Katja Brüeckner, University of California, San Francisco, USA |
| Genetic reagent (*Drosophila melanogaster*) | Tep4-QF>QUAS-mCherry | *Girard et al., 2021* | N/A | Gift from Dr. Utpal Banerjee, University of California, Los Angeles, USA |
| Genetic reagent (*Drosophila melanogaster*) | Ubi-FUCCI | Bloomington *Drosophila* Stock Center | RRID: BDSC_55124 | |

*Continued on next page*

*Continued*

| Reagent type (species) or resource | Designation | Source or reference | Identifiers | Additional information |
|---|---|---|---|---|
| Genetic reagent (*Drosophila melanogaster*) | *UAS-FUCCI* | Bloomington *Drosophila* Stock Center | RRID: BDSC_55117 | |
| Genetic reagent (*Drosophila melanogaster*) | *w1118* | Bloomington *Drosophila* Stock Center | RRID: BDSC_3605 | |
| Antibody | Mouse monoclonal anti-phospho-Histone H3 | Invitrogen | Cat# MA3-064, RRID: AB_2633021 | Used in 1:1000 |
| Antibody | Mouse monoclonal anti-LacZ | Developmental State Hybridoma Bank | Cat# 40–1 a, RRID: AB_2314509 | Used in 1:100 |
| Antibody | Donkey polyclonal anti-mouse Cy5 | Jackson Immunoresearch laboratories Inc | Code: 715-175-151, RRID: AB_2340820 | Used in 1:400 |
| Chemical compound, drug | VECTASHIELD with DAPI | Vector Laboratories | Cat# H-1200, RRID:AB_2336790 | |
| Chemical compound, drug | 16% Paraformaldehyde | ThermoFisher Scientific | Cat#28908 | Used in 4% |
| Chemical compound, drug | Triton X | ThermoFisher Scientific | Cat#BP151100 | Used in 0.1% |
| Chemical compound, drug | Normal Goat Serum | Abcam | Cat# ab7481; RRID:AB_2716553 | Used in 16% |
| Chemical compound, drug | Schneider's *Drosophila* medium | ThermoFisher Scientific | Cat# 21720001 | |
| Chemical compound, drug | Fetal Bovine Serum | ThermoFisher Scientific | Cat# 12483–020 | Used in 15% |
| Chemical compound, drug | Insulin solution from bovine pancreas | Sigma Aldrich | Cat# I0516 | Used in 0.2 mg/mL |
| Chemical compound, drug | Sytox Green | ThermoFisher Scientific | Cat# S7020 | Used in 2 µM |
| Commercial assay or kit | Click-iT EdU kit | Life technologies | Cat# C10337 | See detail protocol in the Methods |
| Software, algorithm | MATLAB | Commercial | https://www.mathworks.com/products/matlab.html | |
| Software, algorithm | FIJI | Source of the software *Schindelin et al., 2012* | https://fiji.sc/ | |
| Software, algorithm | MATLAB script used to create heat maps | Codes deposited in the Tanentzapf lab GitHub (https://github.com/Tanentzapf-Lab/LiveImaging_HematopiesisKinetics_Infection_Ho_Carr; *Ho et al., 2023*) | This study | See the Tanentzapf lab GitHub |
| Software, algorithm | MATLAB scripts used to calculate the number of progenitors, plasmatocyte differentiation, and total number of cells in a LG | Scripts deposited in the study *Khadilkar et al., 2017* | N/A | *Khadilkar et al., 2017* |
| Other | Glass bottom mounting dishes | MatTek Corporation | Cat# P35G-0–14 C | See Immunohistochemistry and antibodies section in the Materials and methods. |
| Other | Incubation system | TOKAI HIT | Cat# INU-ONICS F1 | Temperature set at 25°C. See Long-term ex vivo organ culture and confocal imaging section in the Methods. |

## Resource availability

### Lead contact
Further information and requests for resources and reagents should be directed to and will be fulfilled by the lead contact, Guy Tanentzapf (tanentz@mail.ubc.ca).

### Materials availability
This study did not generate new reagents.

### Data and code availability
All raw data reported in this paper is deposited in the Source Data files in this study. MATLAB scripts used for counting total number of cells in a LG are publicly available (*Khadilkar et al., 2017*). All other custom-written scripts including R and MATLAB scripts used for analyses in this study are available on the Tanentzapf lab GitHub: (https://github.com/Tanentzapf-Lab/LiveImaging_HematopiesisKinetics_Infection_Ho_Carr; copy archived at *Ho et al., 2023*).

## Experimental procedures and subject details

### *Drosophila* stocks
*Drosophila melanogaster* stocks and crosses were maintained on standard cornmeal medium (recipe from Bloomington *Drosophila* Stock Center) in vials or bottles at 25°C. Blood progenitor drivers used were *Tep4-Gal4* (*Avet-Rochex et al., 2010*) and *dome-Gal4* (*Hombría et al., 2005*) (kind gifts from Dr. Lucas Waltzer, Université Clermont Auvergne, France). Blood progenitors were labelled using the following fluorescent markers *UAS-mCD8GFP*, *UAS-dsRed*, or *QUAS-mCherry*. Other lines used were: *eater-dsRed* (*Kroeger et al., 2012*; *Tokusumi et al., 2009*) (kind gift from Dr. Elio Sucena, Instituto Gulbenkian de Ciência, Portugal), *dome-MESO-GFP.nls* (*Oyallon et al., 2016*) (kind gift from Dr. Michele Crozatier, Université de Toulouse, France), *gstD-GFP* (*Sykiotis and Bohmann, 2008*) (kind gift from Dr. Dirk Bohmann, University of Rochester Medical Center, USA), *dome-MESO-lacZ* (*Hombría et al., 2005*) (originally line made by Martin P. Zeidler, Max Planck Institute for Biophysical Chemistry, Germany; kind gift from Dr. Nancy Fossett, University of Maryland, Baltimore, USA), *HmlΔ-dsRed.nls* (*Makhijani et al., 2011*) (kind gift from Dr. Katja Brüeckner, University of California, San Francisco, USA), *Tep4-QF>QUAS-mCherry* (*Girard et al., 2021*) (kind gift from Dr. Utpal Banerjee, University of California, Los Angeles, USA), and *w1118* (G.T), *Ubi-FUCCI* (RRID: BDSC_55124), UAS-FUCCI (RRID: BDSC_55117).

### Immunohistochemistry and antibodies
LGs were dissected in ice cold Phosphate Buffer Saline (PBS). The dissected LGs were fixed in 4% paraformaldehyde (PFA) for 15 min, washed with 0.1% PTX (PBS with 0.1% Triton-X [ThermoFisher Scientific, BP151100]) three times (each 5 min), then blocked with 16% Normal Goat Serum (ab7481, abcam) for 15 min followed by an overnight primary antibody incubation at 4 °C. The samples were washed with 0.1% PTX three times (each 5 min) and then blocked with 16% Normal Goat Serum for 15 min. The LG samples were then incubated in appropriate secondary antibodies for 2 hr at room temperature, followed by washes with 0.1% PTX three times (each 5 min) and then mounted in VECTASHIELD with DAPI (H-1200, Vector Laboratories, RRID:AB_2336790) in the glass bottom mounting dishes (MatTek Corporation, 35 mm, P35G-0–14 C). All the antibodies were diluted in 0.1% PTX. The following primary antibodies were used: mouse anti-phospho-Histone H3 antibody (1:1000, Invitrogen, 6HH3-2C5, MA3-064, RRID: AB_2633021) and mouse anti-LacZ antibody (1:100, DSHB 40–1 a, RRID: AB_2314509). The secondary antibody donkey anti-mouse Cy5 (1:400, Code: 715-175-151, RRID: AB_2340820, Jackson Immunoresearch laboratories Inc) was used.

### Long-term ex vivo organ culture and confocal imaging
Early third instar larvae (84 hr after egg laying [AEL]) were chosen for all long-term live imaging experiments, washed using PBS for three times, followed by a quick rinse with 70% ethanol, then washed again using PBS for three times. Organs including larval LGs, fat bodies, ring gland, central nervous system, and heart tube were dissected in *Drosophila* Schneider's medium (ThermoFisher Scientific, Catalog number 21720001) in room temperature. The connection between CNS, ring gland, LG,

and heart tube should be maintained during all steps from larval dissection, organ mounting and to confocal imaging. The dissected organs were then placed and mounted in the Schneider's medium supplied with 15% FBS (ThermoFisher Scientific, Catalog number 12483–020) and 0.2 mg/mL insulin (Sigma I0516) in a glass bottom dish. The medium was prepared fresh in 10 min prior to dissection in room temperature. The LG was mounted in such a manner to align the dorsal-ventral axis of the tissue with the z-axis of the confocal optical section. To stabilize the LG, the organs were covered with a 1% agar pad and spacers made from 1% agar were placed in between agar pad and glass bottom dish to shield the organs from mechanical force. Optimal moisture conditions during live imaging was maintained by the addition of 2 ml of the medium on top of the agar pad. All live imaging experiments were performed at 25°C in a microscope incubation chamber (TOKAI HIT, Catalog number: INU-ONICS F1). LGs were imaged using an Olympus FV1000 inverted confocal microscopy with a numerical aperture 1.30 UPLFLN 40 X oil immersion lens. Imaging duration varied due to movement caused by occasional heart tube contractions (see *Figure 1B*). The middle two planes of LGs spaced by 1.5 μm were imaged at a 15 seconds interval using lasers with the excitation wavelength at 488 nm (green laser) and 561 nm (red laser). The parameters were chosen to minimize phototoxicity, increase temporal resolution, and maximize the number of cells captured in each experiment. To avoid phototoxicity and photobleaching in the LGs, the laser was kept at 1% power (*Icha et al., 2017*), which is the weakest laser power that provides a good signal for live LGs in the FV1000 confocal microscopy. Using the laser power setting, no noticeable photobleaching (i.e. the signal levels did not drop down substantially over the course of imaging; see multiple videos and time-lapse images in this study as pieces of evidence) or phototoxicity (the main cause of which is an increased ROS level in a sample upon strong laser illumination *Icha et al., 2017*; see *Figure 1E* as an evidence showing the ROS level remained low and stable in LGs during imaging). No correction for photobleaching was performed. Time-lapse recordings of LGs and the resulting t-series images were processed using Fiji (*Schindelin et al., 2012*) and MATLAB software. All fluorescent intensity in this study are mean grey values measured in Fiji.

## EdU proliferation assay on LGs

Click-iT EdU (5-ethynyl-2′-deoxyuridine) imaging kit (Life Technologies, Cat# C10337) was used to perform cell proliferation assay. Larvae were washed using PBS three times, quickly rinsed with 70% ethanol, and then washed and dissected in the Schneider medium in room temperature. The LGs (with CNS, ring gland, heart tube, and fat bodies) were cultured in the Schneider medium (with 15% FBS and 0.2 mg/mL insulin) supplied with EdU solution with the final concentration of 10 μM for an hour. Following incubation in the EdU solution, the LGs were fixed in 4% PFA for 15 min, rinsed with 16% Normal Goat Serum twice, washed with 0.1% PTX for 20 min, and then incubated in a Click-iT reaction cocktail (430 μl 1xClick iT reaction buffer, 20 μl CuSO$_4$, 1.2 μl Alexa Fluor azide, and 50 μl 1xClick iT EdU buffer additive) for 30 min at room temperature in dark. After the incubation, the cocktail solution was removed and the LGs were washed twice using 16% Normal Goat Serum (each 10 minutes) and mounted in VECTASHIELD with DAPI in glass bottom dishes. The EdU signal from the LGs was imaged using a laser with the excitation wavelength at 488 nm. Number of EdU-positive cells per primary lobe were counted manually in Fiji using a Cell counter plugin.

## Cell death monitoring during live imaging

Cell death during long-term live imaging was monitored using the nucleic acid stain Sytox Green (ThermoFisher Scientific, Catalog number S7020). The Sytox Green dye functions as an indicator of dying cells as the dye is impermeable to the plasma membrane of live cells. Dissected LGs were incubated and imaged in the Schneider medium containing 2 μM Sytox Green or PBS as a positive control. A stock solution of Sytox Green (5 mM in DMSO) was prepared and diluted in 1:2500 to a final concentration of 2 μM. The LG was imaged immediately after mounted in Sytox Green-containing medium. Cell death was assessed by counting all the progenitors (shown in *Figure 1F* and *Figure 1—figure supplement 2D*) or in the entire LGs (shown in *Figure 1—figure supplement 2E*). No particular subset of progenitors or portion of a LG was chosen to image.

## Real-time tracking of cell cycle phases during live imaging

To track the cell cycle, a Fly-FUCCI system was used (*Zielke et al., 2014*). The Fly-FUCCI system consists of two major UAS transgenes carrying GFP or RFP-tagged degrons: a UAS-GFP.E2f1.1–230 (the N terminus amino acid 1–230 of E2f1 was fused to GFP) and a UAS-mRFP1.CycB.1–266 (the N terminus amino acid 1–266 of CycB was fused to RFP). E2f1 is degraded by the S phase-dependent ubiquitin ligase CRL4$^{Cdt2}$ while CycB is targeted by the APC/C for proteasomal degradation from mid-mitosis throughout G1 phase. By combining the two probes, cells that are in G1 phase are labelled in green (E2f1-GFP accumulation), in S phase are labelled in red (CycB-mRFP accumulation), and in G2 phase are labelled in yellow (presence of both E2f1-GFP and CycB-mRFP). The fluorescent intensities of GFP and RFP of individual cells in the LG in each time point were tracked and exported using the Fiji TrackMate plugin (*Tinevez et al., 2017*) and plotted using GraphPad Prism (Ver. 6).

## Heat map construction

Spatial information of cellular events (including cell division and differentiation) from long-term LG videos was extracted using 2 MATLAB scripts and an image analysis workflow (*Figure 4—figure supplement 2*; Data and Code Availability; Tanentzapf lab GitHub). The workflow contained 9 steps: (1) The frame where cellular events were identified was saved as an image (in.tiff) using Fiji. The heart tube (as a landmark structure) was annotated based on the well-defined location of it with respect to the two lobes of the LG (see step 1 in *Figure 4—figure supplement 2*, heart tube was highlighted in a white line next to the LG lobe). The image was then loaded into the first custom written script (Data and Code Availability; Tanentzapf lab GitHub). (2) The image was rotated to align the heart tube along the y-axis so that the heart tube was in parallel to the y-axis. (3) For later comparison, the image was then flipped so that the lobe was facing the right side and the heart tube was facing the left side. The step was designed to adjust all LG lobes facing the same direction with respect to the landmark structure. (4) The boundary of the lobe and the location where a cellular event was observed were manually selected. (5) A bounding box was created by the script and the width and height of the bounding box were defined. (6) The total width and height of the bounding box were divided equally into five segments (each as 20% of the total width and height, respectively) to create a grid. (7) A single heatmap showing the location of a cellular event was created. (8–9) Multiple heatmaps from different videos were combined as a final heatmap using the second custom script (Data and Code Availability; Tanentzapf lab GitHub). To statistically compare heat maps, correlation coefficients between heat maps were calculated using a corrcoef function in MATLAB. A correlation coefficient value from 0 to 0.25, 0.25–0.5, 0.5–0.75, and 0.75–1 was defined as no correlation, weak correlation, moderate correlation, and strong correlation, respectively. A weak correlation suggested that a shift but not a complete relocation of cellular events was observed.

## Spatiotemporal analyses of progenitor mitotic events

Mitotic events were tracked in blood progenitors labelled by dome-Gal4 >UAS mGFP or dome-MESO-GFP in long-term LG videos using Fiji. The following quantitative analyses were performed on the mitotic events: (1) Duration of mitotic events was defined as the time a mother progenitor spent from the onset of mitosis throughout to the end where the nucleus of two progenies reformed and were clearly visualized (approximately in telophase; *Video 1*; *Video 4*; *Figure 2A–B*). The onset of mitosis was defined as 40 frames (roughly 10 min) before the nucleus breakdown (which happens in prophase) was observed. The same criteria were applied to all mitosis analysis in our study. (2) The cell size of individual daughter cells post-mitosis was tracked over 3 hr and measured in Fiji using a Polygon ROI Selection tool. The ROI was drawn along the cell membrane marked by dome-Gal4 driven membranous GFP. A z stack with 2 slices was projected in Fiji using maximum projection before the measurement. (3) The position of a contractile ring was inferred based on the location where the cleavage furrow occurred in dividing cells. The distance of the contractile ring to the two poles of a dividing cell was measured in Fiji using a Straight Line ROI Selection tool. A z stack with 2 slices was projected in Fiji using maximum projection before the measurement. (4) The JAK-STAT signaling activity (reflected by dome-MESO-GFP intensity) of daughter cells were measured in Fiji using a Circle ROI Selection tool. (5) $\rho$-mitosis: as illustrated in *Figure 3F*, a $\rho$-mitosis was defined as a mitosis occurs on the plane formed by right-left and posterior-anterior axes. Progenitors divide away or towards heart tube on this plane at any angles ranging from 0 to 180 degree are classified as $\rho$-mitosis. (6) z-mitosis: as

illustrated in *Figure 3F*, a z-mitosis was defined as a mitosis occurs along the dorsal-ventral axis at any angles ranging from 0 to 180 degree. (7) The orientation of a $\rho$-mitosis relative to the heart tube was determined based on the angle between the mitosis direction (the direction that was perpendicular to cleavage furrow and parallel to the positions of two newly formed nuclei of daughter cells) and the heart tube. The newly formed nuclei of daughter cells were used to determine the relative position of the cells and infer the plane of division. The angle was manually measured in Fiji using the Angle tool function. To test if the orientation of mitosis follows normal distribution or shows a bias towards a certain direction in the LGs, a Q-Q plot statistical analysis was performed in R. The orientation data from individual progenitors undergoing mitosis were loaded into R and the Q-Q plot was produced using a qqnorm function in the R Stats package. To add a theoretical Q-Q line onto the plot, the QQline function was used. The linearity of the points lining along the Q-Q line suggests that the data follows normal distribution. (8) Distances of $\rho$- and z-mitosis to the heart tube and posterior end of the LG on the heart tube (a well-defined position where the PSC localized) was measured manually in Fiji using the Straight Line ROI Selection tool. (9) The mitotic index of progenitors was calculated by dividing the number of pH3 labelled progenitors (pH3$^+$ dome$^+$) by the total number of progenitors (dome$^+$) in the LG. Number of pH3$^+$ progenitors were counted in Fiji using the Cell Counter plugin. (10) The positions where blood progenitors divide inside a LG were recorded and the information from multiple long-term videos was then summarized in a heat map (as described above in the Heat map construction section) to visualize regions with different level of mitotic activities.

## Information of main markers used to track differentiation

To track blood progenitor differentiations in real-time, dome-MESO-GFP and eater-dsRed were used in combination throughout the study to indicate the differentiation status of a cell (*Figure 2— figure supplement 1B–C*). Pieces of critical information of the two markers were provided. (1) dome-MESO-GFP: The dome-MESO-GFP line is a marker of JAK-STAT positive blood progenitors and was generated by swapping the LacZ part of dome-MESO-LacZ line with a GFP (*Oyallon et al., 2016*; *Hombría et al., 2005*). The construct of the dome-MESO-LacZ transgene was not a complete enhancer trap containing the entire *domeless* gene promoter sequence but rather a 2.8 kb fragment from the first exon and first intron, containing multiple STAT binding sites. The expression of dome-MESO construct has been further shown to be dependent on JAK-STAT signaling, demonstrating that JAK-STAT signaling forms a positive feedback loop where the activity of itself can promote the expression of its own receptor Dome (*Hombría et al., 2005*). This indicates, together with the original study where the dome-MESO-LacZ was developed, that the dome-MESO-GFP/LacZ lines are reliable JAK-STAT signaling reporters to track progenitor cell fate. (2) eater-dsRed: The eater-dsRed line was chosen to track the differentiation status of a progenitor for the following reasons: (a) It was an enhancer-trap line made and verified to be able to accurately reflect spatial-temporal expression of the *eater* gene in the LG (*Kroeger et al., 2012*). (b) A further study confirmed that eater-dsRed marks both differentiated plasmatocytes (high eater-dsRed level) and distal progenitors that already commit to a plasmatocyte fate (lower eater-dsRed level than mature plasmatocytes *Blanco-Obregon et al., 2020*). By tracking eater-dsRed level in combination with dome-MESO-GFP, the full range of the transition of a cell undergoing differentiation can be captured (see *Figure 5* and *Figure 7I–N* as examples). (c) The reason of choosing eater-dsRed instead of HmlΔ-dsRed to track differentiation events was because, as verified and demonstrated in other two studies using genetics and single cell sequencing approaches, the HmlΔ-dsRed line marks both plasmatocytes and crystal cell precursors (Hml$^+$ Lozenge$^+$, summarized in *Figure 2—figure supplement 1B–C*; *Blanco-Obregon et al., 2020*; *Girard et al., 2021*). Thus, using HmlΔ-dsRed as a differentiation marker brings up the possibility of mixing up crystal cells and/or plasmatocytes when cells were tracked in live imaging experiments, which can make analyzing and interpreting data of differentiation kinetics complicated.

## Spatiotemporal analyses of differentiation

Differentiation events were tracked in videos of LGs carrying dome-MESO-GFP (a JAK-STAT signaling activity reporter that marks blood progenitors *Oyallon et al., 2016*; *Krzemień et al., 2007*) and eater-dsRed (a marker that starts to appear from distal committed progenitors to mature plasmatocytes *Kroeger et al., 2012*; *Tokusumi et al., 2009*). Blood progenitors stay in a steady state (cells expressing either dome$^{high}$ eater$^{low}$ before sigmoid differentiation or dome$^{high}$ eater$^{high}$ before linear

differentiation) prior to the beginning of changes in dome and/or eater levels following differentiation. The time point where we can record such changes was therefore the exit point from the steady state and was denoted as 'frame 0 or 0 hr'. To quantify differentiation, the videos were saved as RGB stacks for the Fiji TrackMate plugin (*Tinevez et al., 2017*). All pixel intensities were preserved equivalent as original data without adjustments on brightness and contrast. A tracking function implanted in Track-Mate toolbox with a LoG (Laplacian of Gaussian) detector was applied to follow differentiating blood progenitors in a video. The raw intensity values of GFP and RFP of a cell at individual time points were exported to a spreadsheet (.xml format). The ratio of dsRed:GFP in each cell at individual time points was then calculated from raw dataset and plotted to visualize the curve shape. The dsRed:GFP ratio over time reflected how the two markers change over time relative to each other and how fast a blood progenitor loses its identity. Sigmoid or linear type of differentiation was categorized based on the shape of the ratio curve (see *Figure 5C–F*). The terms 'sigmoid' and 'linear' were used for descriptive purposes in this study but not mechanic and are interchangeable with a more detailed description: a ratio curve in sigmoidal shape showed that at the beginning cells express dome$^{high}$ eater$^{low}$ following up by slow and fast phases of transition, while a ratio curve in linear shape showed that at the beginning cells express dome$^{high}$ eater$^{high}$ following up by a transition in a consistent rate. To normalize real-time fluorescent signals of dome-EMSO-GFP and eater-dsRed to be able to compare signals across samples/videos, a modified version of fluorescence normalization method for live fly guts was performed (*Martin et al., 2018*). The original normalization method required modifications since it was designed for a situation where one marker gradually changes over time while the other marker does not change over time. In comparison, the current study on LGs was dealing with a scenario where the two markers gradually change over time (dome-MESO-GFP and eater-dsRed) and the differences between the two markers at any time points are required to be preserved after normalization. The signals were normalized as follow: First, the RGB stack of a video was inputted into the Fiji TrackMate plugin to obtain the raw intensities of dome-MESO-GFP and eater-dsRed at individual time points. Second, the raw intensities were imported into MATLAB and normalized using the equations (Norm.G = $(G_t - min (G,R))/(max (G,R) - min (G,R))$; Norm.R = $(R_t - min (G,R))/(max (G,R) - min (G,R))$) where the difference between the fluorescent values at every time point ($G_t$ and $R_t$ representing dome-MESO-GFP and eater-dsRed, respectively) and the minimum fluorescent value was divided by the difference between the maximum fluorescent value and the minimum fluorescent value. Minimum and maximum fluorescent values were obtained across the two markers, as shown by the 'min' and 'max' functions of the equation. By using this approach, the patterns, trends, and relative differences between two markers were all preserved to make comparisons across the videos. Moreover, we confirmed that the normalization method preserved the trends of the markers during differentiation (*Figure 7—figure supplement 2A–B*) compared to the raw values before normalization. The custom-written MATLAB code used to perform normalization was deposited in the Tanentzapf lab Github (Data and Code Availability). To quantify the rate of each type of differentiation, a linear regression fit was applied to the fast phase of a sigmoid differentiation curve and to the entire linear differentiation curve using a custom written R script (Data and Code Availability, Tanentzapf lab GitHub). The slope of the fitted line was calculated as follows: Slope = Changes of dsRed:GFP ratio/Time. To analyse the spatial distribution of differentiations in the LGs, a heat map was constructed (as described above in the Heat map construction section). The locations where blood progenitors differentiate inside a LG were recorded and the information from multiple long-term videos was then summarized in a heat map to visualize hot and cold spots of differentiation events.

## In vivo analysis of LGs

To perform in vivo analysis on LGs, a method of histo-cytometry (*Stoltzfus et al., 2020*) was applied to extract fluorescent signals of dome-MESO-GFP and eater-dsRed and the positional information of individual cells from the LGs of wild-type control and *E. coli* infected larvae using the TrackMate plugin. Three main steps of histo-cytometry were performed: (1) Imaging: Entire LGs of genotype dome-MESO-GFP; eater-dsRed were imaged by a FV1000 microscopy with a step size 1.5 μm using exactly equivalent laser settings across wild-type control and infection groups. Importantly, the TrackMate-based automatic method used to perform histo-cytometry analysis works best on single sections with cells clearly separated from each other. To unbiasedly select slices across samples and different groups, we took the slides that are located at 25%, 50%, and 75% of the total thickness (or

z axis) of the LG. The imaged LG slides were saved as OIB files in Fluoview and inputted into Fiji as RGB stacks (in.tiff) for the following analysis. (2) Segmentation: To segment individual nuclei in the imaged LGs, the automatic ROI selection tool implanted in the TrackMate plugin with a LoG detector was used. The Blob diameter was set as 13 and the threshold was set as 2.5 in TrackMate across all LG images to reliably select all nuclei. Using the method, in total 2946 cells (LG#1: 795 cells in total, LG#2: 923 cells in total, LG#3: 1228 cells in total) from the wild type LGs and 2985 cells from the LGs following infection (LG#1: 944 cells in total, LG#2: 871 cells in total, LG#3: 1170 cells in total) were captured. From these cells, an Excel-based method (using the rand() function) was performed to completely randomize their order and then took 2500 cells randomly from the two groups that were used for the *Figure 7I–N* (see *Figure 7—source data 1* file). (3) Visualization: The fluorescent intensities of dome-MESO-GFP and eater-dsRed of individual nuclei were extracted and plotted as a scatter plot using R to visualize the distribution of cells based on the expression levels of dome-MESO-GFP and eater-dsRed markers.

## Oxidative stress measurement in ex vivo LGs

A gstD-GFP reporter line (*Sykiotis and Bohmann, 2008*) was used to measure oxidative stress in ex vivo cultured LGs over 13 hr. GstD-GFP is a sensor designed to detect ROS levels in live tissues/animals and is compatible with live imaging experiments. The GFP intensity from individual lobes of LGs was tracked in Fiji using the Polygon ROI Selection tool over 13 hr. The mean grey value of gstD-GFP intensity was measured. The obtained gstD-GFP fluorescent intensities at individual time points were normalized to the total ROI or primary lobe area ($\mu m^2$) and plotted in GraphPad Prism.

## Larval bacterial infection

Ampicillin-resistant, GFP expressing *E. coli* (kind gift from Dr. Christopher Loewen, University of British Columbia, Canada) was used in this study. *E. coli* was grown in the LB medium (10 g Bacto tryptone, 5 g yeast extract, and 5 g NaCl were used to prepared 1 L LB medium) overnight in 37°C for infection experiments. A larval oral infection protocol was applied with slight custom modifications (*Khadilkar et al., 2017*; *Siva-Jothy et al., 2018*). Larvae were collected and starved in a vial containing only 1% agar for 2 hr in room temperature. Post-starvation the larvae were moved into vials containing either regular fly food with LB medium (as a control, 5 g fly food mixing with 1 ml LB, 8–10 larvae per vial) or regular fly food with *E. coli* culture (as an infection group, 5 g fly food mixing with 1 ml LB containing *E. coli*, 8–10 larvae per vial) for 6 hr in 25°C. Larvae at 78 AEL were infected for 6 hr and dissected at 84 AEL for ex vivo live imaging. The infected larvae were first screened under a fluorescent stereo microscopy (Model: MAA-03/B, serial number: 06.07/07) to confirm that they ingested a large amount of GFP expressing *E. coli* (with clear GFP visualized in the intestine region). The larvae were then used for long-term imaging experiments or in vivo analysis.

## Statistical methods

Statistics were performed using GraphPad Prism (ver. 6). p Values were determined using statistical tests that were detailed in all figure legends and Source data files. The sample size of each analysis was indicated in figure legends. No statistical method was performed to pre-determine sample size.

## Acknowledgements

The authors acknowledge the Bloomington and VDRC *Drosophila* Stock Center, and the DSHB hybridoma bank for fly stocks and antibodies. The authors thank the following individuals for fly stocks, reagents, bacteria stains, and protocols: Dr. Lucas Waltzer, Dr. Michele Crozatier, Dr. Elio Sucena, Dr. Christopher Loewen, Dr. Dirk Bohmann, Dr. Nancy Fossett, Dr. Katja Brüeckner, Dr. Utpal Banerjee, and Dr. Lucy O'Brien. The authors thank the Tanentzapf laboratory for insightful discussions. Funding for this study was provided by the grant to GT from Canadian Institutes of Health Research (Project Grant PJT-156277). KYLH was supported by a 4 Year Doctoral Fellowship from the University of British Columbia.

## Additional information

### Funding

| Funder | Grant reference number | Author |
|---|---|---|
| Canadian Institutes of Health Research | PJT-156277 | Guy Tanentzapf |
| University of British Columbia | 4-Year Doctoral Fellowship | Kevin Yueh Lin Ho |

The funders had no role in study design, data collection and interpretation, or the decision to submit the work for publication.

### Author contributions

Kevin Yueh Lin Ho, Conceptualization, Resources, Formal analysis, Validation, Investigation, Visualization, Methodology, Writing – original draft, Writing – review and editing; Rosalyn Leigh Carr, Data curation, Software, Formal analysis, Visualization, Methodology, Writing – review and editing; Alexandra Dmitria Dvoskin, Formal analysis, Validation, Investigation, Writing – review and editing; Guy Tanentzapf, Conceptualization, Supervision, Funding acquisition, Writing – original draft, Project administration, Writing – review and editing

### Author ORCIDs

Kevin Yueh Lin Ho ⬤ http://orcid.org/0000-0001-8083-7043
Rosalyn Leigh Carr ⬤ http://orcid.org/0000-0003-4371-0881
Guy Tanentzapf ⬤ http://orcid.org/0000-0002-2443-233X

### Decision letter and Author response

Decision letter https://doi.org/10.7554/eLife.84085.sa1
Author response https://doi.org/10.7554/eLife.84085.sa2

## Additional files

### Supplementary files

• MDAR checklist

### Data availability

All data generated or analysed during this study are included in the manuscript and supporting files. All the scripts and software generated has been deposited to the Tanentzapf lab Github (copy archived at *Ho et al., 2023*).

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
