## [Editor Report]

This study represents an important technical advancement in the live-imaging and analytical approaches to the *Drosophila* larval hematopoietic organ called the lymph gland. The new method allows tracking the proliferation and differentiation of progenitor cells ex vivo and provides insights into the modes of differentiation during development and infection. The evidence supporting this is convincing but would be further strengthened if the authors explain the disparity between the speed and proportion of two modes of differentiation as the reviewers suggested.

---

## [Decision Letter]

**Decision letter after peer review:**

Thank you for submitting your article "Kinetics of blood cell differentiation during hematopoiesis revealed by quantitative long-term live imaging" for consideration by *eLife*. Your article has been reviewed by 3 peer reviewers, one of whom is a member of our Board of Reviewing Editors, and the evaluation has been overseen by Utpal Banerjee as the Senior Editor. The reviewers have opted to remain anonymous.

Essential revisions:

1. Substantiate statistical analysis.

Since this study is based on the analysis of live imaging, detailed data analysis, interpretation, and presentation are all core to the paper. All the reviewers raised questions regarding data analysis/presentation or statistics. Please find Rev#1 points 5-6, Rev#2 points 1,2,4,5,6, and Rev#3 points 3,6.

2. Provide additional information about the lymph gland development following ex vivo culture.

Rev#1 and 3 raised questions about how relevant the lymph gland development/response is compared to in vivo status after ex vivo culture. How similar the lymph gland is after ex vivo culture in terms of their growth? How faithful ex vivo culture is in mimicking in vivo conditions after infection? Details can be found in Rev#1 points 2-3 and Rev#3 points 1-2.

3. Validate the two modes of differentiation.

Rev#1 and Rev#2 raised questions as to whether the two modes both give rise to the true mature blood cell. Additional spatial and temporal analysis on progenitors or mature blood cells originating from two modes needs to be performed. For example, do both modes lead to the differentiation of P1+ true mature blood cells during development or upon infection? Please find details from Rev#1 point 4.

4. Add essential references. Please find Rev#3 point 4.

*Reviewer #1 (Recommendations for the authors):*

Despite the significant advancement made in this study, there are several points to be clarified before publication in *eLife*.

1. It could be a trivial issue, but it was hard for the reviewer to follow the main and supplementary figures along the text as figures are not arranged linearly and some figure callings are even missing in the text.

2. For the infection experiment; although the authors showed that oral infection induces the differentiation of P1+ plasmatocytes or Hnt+ crystal cells (Khadilkar et al., 2017), it is unclear at which time point they dissected the lymph gland following infection and how much the lymph gland was differentiated at the point of observation (Figure 4-7). Lymph glands in vivo would receive systemic immune factors over the immune activation while lymph glands taken out would be devoid of such signals. Thus, differentiation phenotypes of the lymph gland could differ depending on how long the larvae were exposed to a systemic in vivo environment versus an ex vivo culture system. For example, lymph glands being dissected after 6 hours post-infection may show a different differentiation program when compared to lymph glands dissected after 12 or 24 hours post-infection.

3. Concerning the above point, how is the ex vivo culture system biologically relevant following infection, and how close is it compared to lymph glands in vivo? Do lymph glands cultured ex vivo activate immune-responsive genes or representative immune markers comparable to lymph glands in vivo following infection?

4. The authors proposed two different modes of differentiation, sigmoidal and linear. This is a compelling finding and provides interesting insights; however, it is unclear when and where the two modes of differentiation occur during lymph gland development. Given that the sigmoidal mode initiates with Dome_high & eater_low progenitors, the sigmoid mode of differentiation seems to appear in the typical medullary zone. Dome_high abruptly drops to Dome_low and eater_low rises at the end but the level of eater is still low compared to the starting point of the linear mode. However, the linear mode starts with Dome_high & eater_high population, which could indicate the intermediate progenitors. In fact, the authors showed that the linear mode is abundant in the prospective intermediate region (Fig6F) while the ratio of sigmoid mode is higher near the heart (Fig6D). Therefore, the two different modes of differentiation may not go parallel but are consecutive. The authors describe the two modes as if both modes take place to finally differentiate progenitors into mature hemocytes, but this may not be the case. Temporal ambiguity of the two modes hinders the significance of this finding and the authors need to clarify whether the two modes occur in parallel within the same region of the lymph gland.

5. The authors showed that the sigmoid type differentiation induces the eater and turns off the dome (Figure 5C-D). But absolute GFP/RFP intensities changed in this process are rather marginal in Fig7A compared to Figure 5C-D or Fig7A'. A similar discrepancy is seen in Fig7C-D and FigC'-D'. The infection seems to cause a significant increase in eater-RFP in Fig7D' but the absolute value change in Fig7D shows eater rises from 25 to 35ish. Thus, it is not clear how the "norm intensity" reliably represents the absolute value of GFP and RFP in this presentation. How do cells used in the analyses look? Which graph matches better with the real expressions? And do they divide?

6. Statistics are missing in some graphs. In Fig7N, each data points need to be presented rather than showing dynamite plots. Fig1F represents a small sample size and would be better showing each data point rather than a mean value.

*Reviewer #2 (Recommendations for the authors):*

1. The explanation in line 431 seems incorrect, as the increased number of high high cells upon infection is equivalent to the decreased number of low low cells, and thus would not lead to an increase in the absolute number of differentiated cells. upon infection

See my hypothesis in the public review; perhaps the analysis has missed that cells remain in the intermediate state in the sigmoidal path for a very long time.

2. Figure 5C-D. Not clear to me what the scatter represents.

Are these multiple different cells from the same lymph gland tracked over time?

3. Figure 6C-F, could the authors show some sort of actual graph of the LG to indicate where these changes in the division rate are?

This isn't obvious to the non-initiate if this is at the edge, or where.

4. Figure 7A', "-D', " should be plotted with the same x-axis time course to be able to compare the dynamics from one to the other.

5. Need higher n in Figure 7F.

6. Please show how many cells are in the P category for both conditions in Figure 7N.

*Reviewer #3 (Recommendations for the authors):*

To strengthen the science:

1. The authors should compare ex vivo GstD-GFP and Sytox levels to those seen in fixed samples (in vivo), similar to what was done with EdU and cell cycle in Figure S3. It is important to know whether this ex vivo system maintains similar ROS and cell death levels as that seen in vivo to help determine the limitations of the ex vivo co-culture system.

a. For example, based on past studies of ROS in the LG (Ex: Owusu-Ansah, et al. 2009), one would expect to see higher GstD-GFP levels in the progenitors/MZ than in differentiated cells/CZ. But in Figure S2C, that doesn't seem to be the case. ROS levels seem to be low throughout the LG. Please explain.

2. The authors should also compare the numbers/ratios of progenitors and differentiated cells at the beginning and end of live imaging with in vivo LGs from age-matched larvae. This would help determine whether the process of differentiation they follow ex vivo produces similar results as that seen in vivo.

a. While this is beyond the scope of the current paper, staining or imaging of key molecules that are expected to be high in progenitors (E-cad, Ci, etc) or differentiated cells (P1, lz, etc) and comparing levels in vivo vs ex vivo would be helpful validation of the system.

3. Correlation coefficient analysis is inconsistent. In the methods, the authors say 0 to 0.5 are defined as a "weak correlation". 0 to 0.5 is a wide range of correlation and is not consistent with standard statistical practices that categorize 0 to 0.25 as not correlated, 0.25 to 0.5 as weak correlation, 0.5 to 0.75 as moderate correlation, and 0.75 to 1 as strong correlation.

a. In fact, on lines 385-395, a correlation coefficient of 0.14 is interpreted as "not greatly altered", while 0.45 is interpreted as "marked differences", and 0.49 as "a general increase".

Considering a correlation coefficient below 0.25 as not correlated is more in line with standard practices but not consistent with the authors' explanation in the methods.

i. The authors should update their methods to reflect a more standard, and consistent, interpretation of correlation coefficients. If there is reason to believe this standard does not apply here, please explain.

b. Furthermore, in lines 311-314, the authors say that a correlation coefficient of 0.44 is consistent with a "weak correlation". There is no substantial difference between 0.44, 0.45, and 0.49, therefore either all should be considered "weak correlations" or all are correlated, the authors should not use or imply different terms for these very similar values. Please update the language to reflect a more consistent interpretation of correlation coefficients.

4. In line 463, the authors say that Hml (but not Pxn) is a mature blood cell marker, and yet both this paper (Figures 4, S1, and S4), as well as Spratford et al. 2021 and Girard et al. 2021, shows Hml is an intermediate progenitor marker and therefore not a mature cell marker. It is curious that these papers are not mentioned when mentioning other papers on intermediate progenitors (e.g. lines 87-90 and lines 458-460) as they present compelling evidence that Hml/dome-meso marks intermediate progenitors (as the authors show in Figures 4/S4) and that IPs serve as a transitional state between progenitors and differentiated cells.

5. One caveat not noted in the manuscript is that because the authors use Eater-Dsred to monitor differentiation they are only able to follow differentiation events leading to plasmatocytes and not crystal cells (although the authors do point out that crystal cells are not thought to transition through an Eater positive transitional state).

6. Some statistics seem to be missing and it is not clear if statistical tests were not performed or if they were performed but found to be not significant. Several conclusions and claims that the authors state (especially in the WT vs infection section) seem to be based on data without statistics and so should either be interpreted with that in mind or it should be explained why the stats could not be performed if this is true. Examples include:

a. Figure 6B, is there any significant difference between linear diff in WT vs infection or sigmoid diff in WT vs infection?

b. Figure 7E, are there any significant differences between WT and infection in those graphs?

i. For example, whether these changes are significantly different affect the authors' conclusions in lines 413-416

c. Figure 7N, are there any significant differences between WT and infection in those graphs? (be careful here how sample size (n=) is defined because each individual lobe can be considered a replicate but not each cell.)

d. Figure 7I-N, why are there so few cells? Were the number of cells in each lobe counted in 3D? If so, there should be many more cells total for 8 lobes. Please explain.

e. Actual p-values should be included in the figure legends (stars can be kept in the figures themselves).

---

## [Author Response]

Reviewer #1 (Recommendations for the authors):Despite the significant advancement made in this study, there are several points to be clarified before publication in eLife.1. It could be a trivial issue, but it was hard for the reviewer to follow the main and supplementary figures along the text as figures are not arranged linearly and some figure callings are even missing in the text.

Thank you for pointing this out. We have gone through the manuscript systematically to ensure a linear order of figure callouts and make sure all figure callouts are where they are supposed to be.

2. For the infection experiment; although the authors showed that oral infection induces the differentiation of P1+ plasmatocytes or Hnt+ crystal cells (Khadilkar et al., 2017), it is unclear at which time point they dissected the lymph gland following infection and how much the lymph gland was differentiated at the point of observation (Figure 4-7). Lymph glands in vivo would receive systemic immune factors over the immune activation while lymph glands taken out would be devoid of such signals. Thus, differentiation phenotypes of the lymph gland could differ depending on how long the larvae were exposed to a systemic in vivo environment versus an ex vivo culture system. For example, lymph glands being dissected after 6 hours post-infection may show a different differentiation program when compared to lymph glands dissected after 12 or 24 hours post-infection.

The reviewer raises two excellent points, ones we have thought about a lot and took into consideration. First, we would like to state that reading the reviewer’s comments here, as well as other reviewer comments, convinced us we needed to add a paragraph to the discussion spelling out the possible limitations of the ex vivo culture method. This new paragraph is the second paragraph of the Discussion line 493-503 in the revised manuscript and directly brings up the reviewer concerns expressed above. Specifically, it is difficult to prove beyond doubt that differentiation is identical following infection in cultured, ex vivo*,* conditions and in vivo conditions. However, we think our data does address this issue at least partially as outlined below.

a) The reviewer asks about the persistence of systemic signals after dissection and start of the ex vivo culture procedure following infection versus in vivo LGs. In particular the reviewer is concerned that lymph glands at 6 hrs post-infection may show a different differentiation program compared to lymph glands at 12 or 24 hours post-infection. We agree this is a legitimate concern, that points to one of the limitations of the kind of ex vivo culture protocols we use here. We attempted to address this issue by using the infection protocol on the same population of larva and then splitting them into a group that was analyzed in vivo (with flies dissected at set intervals and lymph glands fixed, stained , and analyzed), and a group that was analyzed with the ex vivo culture protocol. The results are presented in Figure 6 —figure supplement 1. The in vivo data is somewhat more variable but we do not detect statistically significant differences between the two groups. This data is shown now discussed in the Results line 405-407.

b) The paper includes several other comparisons of the behavior of ex vivo and in vivo models following infection. As far as we could see, within the limit of looking at dynamic data ex vivo versus static data for the in vivo, the two groups behaved quite similarly. For example, when we looked at proliferation (Figure 4—figure supplement 1 in vivo data versus Figure 4H ex vivo data), the proportion of blood progenitors undergoing mitosis decreased in a similar manner both in vivo (assayed using pH3) and ex vivo LGs (assayed using live imaging). Another example, is looking at differentiation, between the two groups we see similar trends in terms of number of differentiation events (Figure 7I-N in vivo data versus Figure 6A ex vivo data), and type of differentiation events (Figure 7I-N in vivo data versus Figure 6B ex vivo data). We now created a new section in Discussion line 493-500 to highlight potential technical limitations such as this and our efforts to mitigate them.

c) To answer the reviewer’s question we started sustained oral infection 78 hours after egg laying, and dissected at 84 hours after egg laying. This is a shorter infection protocol than the one we used previously (in Khadilkar et al., 2017) where infection lasted for 12 hours. This information is outlined in the Methods section line 1096-1097.

3. Concerning the above point, how is the ex vivo culture system biologically relevant following infection, and how close is it compared to lymph glands in vivo? Do lymph glands cultured ex vivo activate immune-responsive genes or representative immune markers comparable to lymph glands in vivo following infection?

As we highlight in the reply to point 2 above we observe similarities in the in vivo and ex vivo response following infection. The additional question asked by the reviewer here, whether we see an elevation of immune markers in the ex vivo culture is an intriguing one. We believe some of our data already provides answers. Specifically, we employed the eater-dsRed enhancer trap line to visualize plasmatocytes. The gene *eater* encodes the protein that is essential for plasmatocytes to phagocytose bacteria (Kocks C. et al., *Cell*, 2005). Eater itself has also been used previously as a readout of immune responsive gene or an immune marker (for example see Ye Y.H., McGraw E.A, *BMC Res Notes*, 2011). It has been shown that the expression of *eater* gene is transcriptionally regulated (Kroeger, P. T., JR. et al., *Genesis*, 2012) and that following infection, the expression of *eater* was upregulated (Ye Y.H., McGraw E.A, *BMC Res Notes*, 2011). If the reviewer looks at Figure 7J (not infected) versus Figure 7L (infected) they will notice a general shift of the blood cells to higher levels of eater-dsRed expression in essentially all the cell types. This data is by no means conclusive but is suggestive of immune activation. We now mention this in the Results section line 459-461.

4. The authors proposed two different modes of differentiation, sigmoidal and linear. This is a compelling finding and provides interesting insights; however, it is unclear when and where the two modes of differentiation occur during lymph gland development. Given that the sigmoidal mode initiates with Dome_high & eater_low progenitors, the sigmoid mode of differentiation seems to appear in the typical medullary zone. Dome_high abruptly drops to Dome_low and eater_low rises at the end but the level of eater is still low compared to the starting point of the linear mode. However, the linear mode starts with Dome_high & eater_high population, which could indicate the intermediate progenitors. In fact, the authors showed that the linear mode is abundant in the prospective intermediate region (Fig6F) while the ratio of sigmoid mode is higher near the heart (Fig6D).

Our thinking is very much in line with the reviewer’s suggestion that the cell population we identify as undergoing the linear trajectory overlaps with the population previously identified as “intermediate progenitors''. This is based on the location where they differentiate and the profile of markers they express. However we did not state this outright out of abundance of caution. The population of “intermediate progenitors” as described in Blanco-Obregon et al., 2020 *Developmental Biology* and Spratford et al., 2021 *Development* papers, are defined as expressing both the mature hemocyte marker Hml and the progenitor marker Dome. In our study, we used Dome and *eater*, not Dome and Hml. We used *eater* and not Hml is because Hml also marks crystal cells (Figure 2—figure supplement 1C, as discussed in the section “Information of main markers used to track differentiation” in our Methods section line 994-1000 based on Blanco-Obregon et al., 2020 and Girard et al., 2021). It is possible we were cautious to a fault here so we have now added in the Discussion line 527-529 to mention that cells undergoing linear type differentiation are likely to be intermediate progenitors.

Therefore, the two different modes of differentiation may not go parallel but are consecutive. The authors describe the two modes as if both modes take place to finally differentiate progenitors into mature hemocytes, but this may not be the case. Temporal ambiguity of the two modes hinders the significance of this finding and the authors need to clarify whether the two modes occur in parallel within the same region of the lymph gland.

We should have mentioned this but we never observed the consecutive mode described by the reviewer in hundreds of hours imaging lymph glands. That is, we did not observe a linear type followed by a sigmoid differentiation or a sigmoid type followed by a linear differentiation. As the reviewer notes “the level of eater is still low compared to the starting point of the linear mode”, therefore our data is not consistent with a consecutive model. Nonetheless, in an attempt to address the “temporal ambiguity” pointed out by the reviewer we decided to specifically show that the two pathways could either occur in parallel or at different time points within the same region of the lymph gland. This analysis of differentiation trajectories in both space and time (which we should mention is extremely time consuming, limiting the possible number of data points) is shown in Figure 6. We searched in our data for areas where we could find cells undergoing both the sigmoid and linear types of differentiation occurring in the same area in lymph glands from control and infected larva. Shown in Figures 6D and 6F (wildtype control) and Figure 6E and 6G (infected larva) are regions where we identified cells undergoing the two types of differentiation trajectories in the same region. We then followed each of these cells individually to determine when the differentiation event took place. We identified the time point where differentiation initiated and represented it as the percentage of time into the movie (that is, 25% means 25% of the entire movie had already been recorded when differentiation started, a similar procedure to that carried out in Figure 1—figure supplement 3F). For example, in the wild type, we show two cells (cell 1 and cell 2) in box#1 that underwent sigmoid and linear type differentiation, respectively, starting at the same time while the cells in the box labeled #2 underwent differentiation that started at different time points. In infection group for example, we saw in box#1 Cell 2 and Cell 3 that underwent sigmoid and linear differentiation, respectively, starting at the same time point while other cells in other boxes all differentiate at different time points. This data shows that parallel differentiation takes place, it of course does not discount the “consecutive model” but as we mention above, we have not directly observed such a trajectory. So in conclusion, to date we clearly see the two modes of differentiation occurring in parallel or at different time points but never observed them occurring consecutively. This new data is shown in Figure 5—figure supplement 1 and discussed in line 394-400 and 420-423 of the Results section.

We now also mention in the Discussion line 508-512 that our data is in line with single cell sequencing data for the LG that has shown that there are multiple distinct paths that a progenitor can differentiate into a plasmatocyte (see Figure 5h in this publication: Cho et al., Single-cell transcriptome maps of myeloid blood cell lineages in *Drosophila*, *Nat Coms*, 2020). Their data suggests that there is a PH4-PM1-PM2 path (which contains an intermediate state during differentiation) and a PH5/6-PM3/4 path (which contains no intermediate state). PH: progenitors; PM: plasmatocytes.

5. The authors showed that the sigmoid type differentiation induces the eater and turns off the dome (Figure 5C-D). But absolute GFP/RFP intensities changed in this process are rather marginal in Fig7A compared to Figure 5C-D or Fig7A'. A similar discrepancy is seen in Fig7C-D and FigC'-D'. The infection seems to cause a significant increase in eater-RFP in Fig7D' but the absolute value change in Fig7D shows eater rises from 25 to 35ish. Thus, it is not clear how the "norm intensity" reliably represents the absolute value of GFP and RFP in this presentation. How do cells used in the analyses look? Which graph matches better with the real expressions? And do they divide?

We thank the reviewer for pointing this out as we should have explained better the data shown in Figure 7 and Figure 5, which would have prevented this misunderstanding. First we wanted to explain why and how we normalized the data shown in Figures 5C-D and in Figure 7C-D. The reason expression needs to be normalized is in order to be able to compare signals across multiple samples and several independent experiments. Our normalization protocol is a modified version of the one established by Martin et al., (2018, see Methods section “Spatiotemporal analyses of differentiation” in line 1024-1041). The original normalization method from Martin et al., 2018 required modifications since it was designed for a situation where one marker gradually changes over time while the other marker does not change over time. In comparison, in our present study on LGs we dealt with a scenario where both markers used (dome-MESO-GFP and eater-dsRed) gradually changed over time.

We have taken the reviewer’s concern to heart and went back to our dataset and re-analysed it to compare normalized and raw data. The results of this analysis are presented in Figure 7—figure supplement 2 and in Methods line 1041-1043, raw data shown in the two bars on the left and normalized data on the two bars on the right. We think this data argues that normalization does not change the overall interpretation.

We have included with the manuscript movies (Video 6-9) corresponding to Figure 7A-7D that illustrate how the cells look and how intensities of dome GFP and eater-dsRed change overtime. The colors used in the Videos S6-S9 was based on raw intensities of dome GFP and eater-dsRed. We now also included the images in Figure 7 —figure supplement 1 showing how a single cell from Figure 7A-7D looked like.

Finally, we did not observe cells undergoing differentiation and mitosis at the same time nor cells divide after they differentiate in all our live imaging videos.

6. Statistics are missing in some graphs. In Fig7N, each data points need to be presented rather than showing dynamite plots. Fig1F represents a small sample size and would be better showing each data point rather than a mean value.

We again apologize for not being clearer about this but the data shown in figure 7N is simply an unprocessed raw number, not an average, from a total cell count of cells from 6 primary lobes of 3 LGs (now mentioned in Methods line 1055-1076). In other words, what is shown is the complete data set for 3 entire lymph glands, specifically the control lymph glands shown in Figure 7J and the lymph glands from infected larvae shown in Figure 7L. This was indicated in Results (line 471) and figure legend (line 780-781) sections. That is, Figure 7N showed the total number of cells (shown individually as dots) in each quadrant of 7J and 7L. The benefit of this (also brought up by the reviewer 3’s point #5) is that showing the numbers of cells in each quadrant helps the reader interpret the data more easily. We made this explicit in the figure legends now.

Also, as the reviewer suggested we modified Figure 1F to show all data points.

Reviewer #2 (Recommendations for the authors):1. The explanation in line 431 seems incorrect, as the increased number of high high cells upon infection is equivalent to the decreased number of low low cells, and thus would not lead to an increase in the absolute number of differentiated cells. upon infectionSee my hypothesis in the public review; perhaps the analysis has missed that cells remain in the intermediate state in the sigmoidal path for a very long time.

We agree with the reviewer that we probably overstated the link between the change in population of the two intermediate populations (dome low eater low and dome high eater high). We have rewrote this section (now in line 467-477) to be descriptive and clearly state the parts that are speculative.

2. Figure 5C-D. Not clear to me what the scatter represents.Are these multiple different cells from the same lymph gland tracked over time?

Sorry for not making this clearer but both Figure 5C and Figure 5D represent a single cell that was tracked over time. In Figure 5C, each dot represents the fluorescent intensity of dome-GFP or eater-dsRed at that time point, while 5D shows the ratio of the green and red signal at that time point. We modified the figure legends of Figure 5C-5D in line 733-734 and line 736-737 to state this in clearer fashion.

3. Figure 6C-F, could the authors show some sort of actual graph of the LG to indicate where these changes in the division rate are?This isn't obvious to the non-initiate if this is at the edge, or where.

To make this clearer to our readers we previously included the following image with the grid we used overlaying on actual lymph gland (the figure is now in Figure 4 —figure supplement 2 #6 panel which described how a heat map was generated; see Methods). Left panel from Figure 4 —figure supplement 2 and right panel from Figure 6C show the relationship between a heatmap and an actual LG. The coordinates of the two graphs are in the same orientation (see the pink dot as a reference point).

4. Figure 7A', "-D', " should be plotted with the same x-axis time course to be able to compare the dynamics from one to the other.

This is a fair request but a bit of a tricky one because the differences in time between the linear and sigmoid trajectory meant that the data looked compressed or elongated when plotted at identical scales in different trajectories making it hard to interpret the data. However, in line with the reviewer we now made the x axis of Figure 7A’-A’’ and 7B’-7B’’ uniform and the x axis in Figure 7C’-7C’’ and 7D’-7D’’ uniform. This makes comparison within trajectories easier to compare yet still allows the shape of the curve to be apparent.

5. Need higher n in Figure 7F.

We want to assure the reviewer that we tried! When compared to a lymph gland from an infected, or immune-activated, larva, wildtype controls do not exhibit a great deal of differentiation at this developmental stage (early third instar larval stage). It took a great deal of effort to get the sample size we presented in the previous version of Figure 7F. Nonetheless, we did our best to increase the sample size for WT linear differentiation and the figure is now in Figure 7G.

6. Please show how many cells are in the P category for both conditions in Figure 7N.

We have added this information into Figure 7N.

Reviewer #3 (Recommendations for the authors):To strengthen the science:1. The authors should compare ex vivo GstD-GFP and Sytox levels to those seen in fixed samples (in vivo), similar to what was done with EdU and cell cycle in Figure S3. It is important to know whether this ex vivo system maintains similar ROS and cell death levels as that seen in vivo to help determine the limitations of the ex vivo co-culture system.

We have added the GstD-GFP experiment the reviewer asked for, and added it now in Figure 1 —figure supplement 2C. Our results show that our ex vivo system maintains similar ROS levels as that seen in vivo. We now mention this data in Results line 160-162.

As for analysis of cell death, we are lucky to have access to many studies that measured the numbers of dying cells in lymph glands in vivo in 3rd instar larvae which we can compare to our ex vivo culture conditions. In ex vivo culture, we showed that following 12.5 hrs of culture period, the average number of dying cells ranged from 2-10 cells (see Author response image 1 which was derived from Figure 1 —figure supplement 2E left panel). This is comparable to 0~10 dying cells reported for wildtype 3^rd^ instar lymph glands (see Figure S1K from Mondal et al., 2011, *Cell* doi.org/10.1016/j.cell.2011.11.041); Figure 1 of Chiu H. and Govind S., 2002, *Cell death & differentiation* (doi: 10.1038/sj.cdd.4401134); Figure 3G of Yu S et al., 2021, *eLife* (doi.org/10.7554/*eLife*.60870); Figure S7A from Khadilkar et al., 2020, *Cur Biol* (doi.org/10.1016/j.cub.2020.06.027); and Figure 6A, 6B, 6C, 6H from Araki et al., 2019, *Diseases Models and Mechanisms* (doi.org/10.1242/dmm.037721). We therefore conclude that the number of dying cells in the ex vivo cultured LGs falls into the normal range that is expected to be observed in in vivo LGs with 12 hours of culture. We now mention this explicitly in the text in Results line 168-172. However, after about 12.5 hours the rate of cell death under the ex vivo culture conditions we begin to see a rate of cell death that is higher than that reported in the controls. Indicating that beyond 13 hours the ex vivo culture is less reliable. This is why we capped all out movies well before we got to this time point.

**Author response image 1. sa2fig1:** 

a. For example, based on past studies of ROS in the LG (Ex: Owusu-Ansah, et al. 2009), one would expect to see higher GstD-GFP levels in the progenitors/MZ than in differentiated cells/CZ. But in Figure S2C, that doesn't seem to be the case. ROS levels seem to be low throughout the LG. Please explain.

The images in the previous Figure S2C (the figure is now in Figure 1 —figure supplement 2A-B) indeed do not display a very obvious difference in the level of gstD-GFP between MZ and CZ. However, it is hard to tell since we did not include the red channel (to image eater-dsRed) that can help to distinguish CZ from MZ or a DIC channel that can help to visualize an entire LG. In images with the DIC channel included, the difference between MZ and CZ (see Author response image 2) is quite apparent. Importantly, the crux of our argument in doing this experiment is that whatever the overall levels of gstD-GFP in the lymph gland are, oxidative stress did not dramatically increase through the course of the live imaging period.

2. The authors should also compare the numbers/ratios of progenitors and differentiated cells at the beginning and end of live imaging with in vivo LGs from age-matched larvae. This would help determine whether the process of differentiation they follow ex vivo produces similar results as that seen in vivo.

As suggested by the reviewer we now included data comparing the ratios of differentiated cells to progenitors at various time points from ex vivo wild-type LGs and in vivo wild-type LGs from age-matched larvae (Figure 1 —figure supplement 3G). Specifically, although slight variation was observed at different time points, which is to be expected in an evolving system such as the lymph gland, differences in the ratio observed between the ex vivo and in vivo conditions were not statistically significant. See line 345-348.

a. While this is beyond the scope of the current paper, staining or imaging of key molecules that are expected to be high in progenitors (E-cad, Ci, etc) or differentiated cells (P1, lz, etc) and comparing levels in vivo vs ex vivo would be helpful validation of the system.

The stainings suggested by the reviewer are all excellent suggestions and we will add these to our to-do list for the future.

3. Correlation coefficient analysis is inconsistent. In the methods, the authors say 0 to 0.5 are defined as a "weak correlation". 0 to 0.5 is a wide range of correlation and is not consistent with standard statistical practices that categorize 0 to 0.25 as not correlated, 0.25 to 0.5 as weak correlation, 0.5 to 0.75 as moderate correlation, and 0.75 to 1 as strong correlation.a. In fact, on lines 385-395, a correlation coefficient of 0.14 is interpreted as "not greatly altered", while 0.45 is interpreted as "marked differences", and 0.49 as "a general increase".Considering a correlation coefficient below 0.25 as not correlated is more in line with standard practices but not consistent with the authors' explanation in the methods.i. The authors should update their methods to reflect a more standard, and consistent, interpretation of correlation coefficients. If there is reason to believe this standard does not apply here, please explain.b. Furthermore, in lines 311-314, the authors say that a correlation coefficient of 0.44 is consistent with a "weak correlation". There is no substantial difference between 0.44, 0.45, and 0.49, therefore either all should be considered "weak correlations" or all are correlated, the authors should not use or imply different terms for these very similar values. Please update the language to reflect a more consistent interpretation of correlation coefficients.

This is a very good point. We needed to have a consistent standard for correlation and we now use the following suggested standard: 0 to 0.25 as not correlated, 0.25 to 0.5 as weak correlation, 0.5 to 0.75 as moderate correlation, and 0.75 to 1 as strong correlation. We modified the text in the Results accordingly as well as explained in the Methods section line 927-931.

4. In line 463, the authors say that Hml (but not Pxn) is a mature blood cell marker, and yet both this paper (Figures 4, S1, and S4), as well as Spratford et al. 2021 and Girard et al. 2021, shows Hml is an intermediate progenitor marker and therefore not a mature cell marker. It is curious that these papers are not mentioned when mentioning other papers on intermediate progenitors (e.g. lines 87-90 and lines 458-460) as they present compelling evidence that Hml/dome-meso marks intermediate progenitors (as the authors show in Figures 4/S4) and that IPs serve as a transitional state between progenitors and differentiated cells.

Thanks for pointing this out, we edited the text in line with the points raised and added the missing citations into line 87-88 and line 512-514 (the previous 458-460).

5. One caveat not noted in the manuscript is that because the authors use Eater-Dsred to monitor differentiation they are only able to follow differentiation events leading to plasmatocytes and not crystal cells (although the authors do point out that crystal cells are not thought to transition through an Eater positive transitional state).

Using our long term live imaging protocol to capture crystal cell differentiation will definitely be an important goal of our future project. The reason why we did not study crystal cells currently is because we did not have a line to label crystal cells in the red channel (the BcF6GFP enhancer trap line which would work is GFP based) which we need to be used together with dome-MESO-GFP. We now inserted a mention of this caveat in Discussion line 500-503.

6. Some statistics seem to be missing and it is not clear if statistical tests were not performed or if they were performed but found to be not significant. Several conclusions and claims that the authors state (especially in the WT vs infection section) seem to be based on data without statistics and so should either be interpreted with that in mind or it should be explained why the stats could not be performed if this is true. Examples include:a. Figure 6B, is there any significant difference between linear diff in WT vs infection or sigmoid diff in WT vs infection?

We now performed the statistical tests requested in Figure 6B. There is a statistically significant difference in the linear differentiation trajectory between wild type LGs and LGs from infected larvae while there is no statistically significant difference in the sigmoid differentiation trajectory between wild type LGs and LGs from infected larvae. We updated Figure 6B to show this. Importantly, this does not impact our interpretation of the data in the Results.

b. Figure 7E, are there any significant differences between WT and infection in those graphs?i. For example, whether these changes are significantly different affect the authors' conclusions in lines 413-416

We now performed the requested statistical analysis on the previous Figure 7F-7H (figures are now in Figure 7E-G). The conclusion now reads “Taken together, the data is consistent with a model whereby infection causes higher differentiation in the LGs not by increasing the rate of differentiation but rather by inducing a shift from one type of differentiation, the sigmoid trajectory, to another, the linear trajectory.” and is in line with the result of the statistical test (now described in line 445-448).

c. Figure 7N, are there any significant differences between WT and infection in those graphs? (be careful here how sample size (n=) is defined because each individual lobe can be considered a replicate but not each cell.)

Figure 7N was designed to show the total number of cells in each quadrant of Figure 7J and 7L. Each bar represents a single value which shows the sum of the number of cells in each quadrant. We now explicitly stated the number of cells as well in each category of wild type and infection in the text where we describe the Results of Figure 7N in line 467-471.

d. Figure 7I-N, why are there so few cells? Were the number of cells in each lobe counted in 3D? If so, there should be many more cells total for 8 lobes. Please explain.

We have to apologize, this is definitely our mistake here, and we thank the reviewer by pointing this out.

First, we made a mistake in the sample size that was listed in the previous version of the graph (however the source data file had the correct information). Specifically, the sample size is 3 LGs that contain 6 primary lobes. We now update sample size on the graph of Figure 7I-N, figure legends (line 774-776), and Results (line 454-455).

Second, the TrackMate-based automatic method we used to perform histo-cytometry analysis can only work on single sections (not a stack) with cells clearly separated from each other. To unbiasedly select slices across samples and different groups, we took the slides that are located at 25%, 50%, and 75% of the total thickness (or z axis) of the LG. That means from each LG, we sampled three slides that are fed into the Fiji TrackMate plugin. Using this method, we took in total 2946 cells (LG#1: 795 cells in total, LG#2: 923 cells in total, LG#3: 1228 cells in total) from the wt LGs and 2985 cells from the LGs of infection (LG#1: 944 cells in total, LG#2: 871 cells in total, LG#3: 1170 cells in total). From these cells, we first performed an Excel-based method (using the rand() function) to completely randomize their order and then took 2500 cells randomly from the two groups that were used for the Figure 7I-N. We now corrected the sample size on the figure and include the above explanation into the Methods section line 1055-1076.

e. Actual p-values should be included in the figure legends (stars can be kept in the figures themselves).

Although all P values were already reported in the Source data files of each figure that we included, we followed the reviewer’s suggestion to put them in the actual figure legends of every figure to make them easier to find.